# Statistical crystallography reveals an allosteric network in SARS-CoV-2 M^pro
Anne Creon[1], T. Emilie S. Scheer [1], Patrick Reinke [1], Aida Rahmani Mashhour [1],
Sebastian Günther [1], Stephan Niebling [4,5], Kira Schamoni-Kast [6,7], Charlotte Uetrecht [6,7],
Alke Meents [1], Henry N. Chapman [1,2,3], Janina Sprenger[1] & Thomas J. Lane [1,2] ✉

To interpret and transmit biological signals, proteins use correlated motions. Experimental determination of these dynamics and the structural distributions they generate remains a key challenge. Here, using 1146 crystal structures of the main protease (M^pro) from SARS-CoV-2, we were able to infer a model of the enzyme's structural fluctuations. M^pro is regulated by concentration, becoming enzymatically active after forming a homodimer. To understand the structural changes that enable dimerization to activate catalysis, we employed our model, predicting which regions of the dimerization domain are structurally correlated with the active site. Mutations at these positions, expected to disrupt catalysis, resulted in a dramatic reduction in activity in one case, a mild effect in the second, and none in the third. Additional crystallography and biophysical experiments provide a mechanistic explanation for these results. Our work suggests that a statistical crystallography, in which numerous crystallographic datasets are analyzed, can reveal the structural fluctuations of protein native states and help uncover their biological function.

The amino acid sequences dictate not only protein structure, but dynamics[1,2]. To understand these dynamics and how they support biological function, new experimental methods that provide an atomic-resolution view of protein structural distributions are needed[3]. Notable efforts in this direction include time-resolved and electric field crystallography[4,5], neural networks that learn structural distributions from single particle cryoEM data[6–10], mass spectrometry methods such as hydrogen-deuterium exchange[11], and NMR techniques such as relaxation-dispersion measurements[12,13]. While powerful, none of these techniques can reveal dynamics—or the structural distributions these dynamics generate—at near-atomic resolution for any general protein or complex of interest. Therefore, methods that expand the range of such studies, especially at high resolution, are of great interest.

One important goal is the ability to routinely identify the correlated atomic fluctuations of native states, where a change in structure in one part of a folded protein results in a corresponding change in a distal region. Such correlations can regulate protein function—most prominently in the case of allostery, where binding of an allosteric effector regulates the function of a distal active site[14]. In the first studies of allostery, the effector was assumed to be a small-molecule ligand. The concepts, however, translate directly to the

situation where the effector is a macromolecule[15], for example, as observed with kinases[16] and DNA binders[17]. Herein, we adopt a broad definition of allostery, including protein complex formation in particular, as a potential effector of allosteric regulation.

The main protease (M^pro) of SARS-CoV-2 provides an important example of an enzyme under such allosteric regulation[18]. Rising to prominence as a drug target during the COVID-19 pandemic[19], M^pro cleaves nascently synthesized viral polyproteins into their constituent protein units and is therefore essential for viral replication. M^pro folds as a monomer but is active only as a dimer, modulating catalysis as a function of concentration. Structural correlations between the active site and dimer interface enable the transformation of M^pro from an inactive form to an active form upon dimerization (Fig. 1)[18,20,21]. Since the active site is complete and folded in the monomer[20], the structural changes that accompany the monomer to dimer transition—and how these activate the enzyme—are a topic of significant interest.

Previous studies of how dimerization activates M^pro have often focused on characterizing mutants of residues that bridge the dimer interface. Such mutants typically eliminate key cross-dimer interactions that generate monomeric species, which are consequently inactive. These studies have

[1]Center for Free-Electron Laser Science CFEL, Deutsches Elektronen-Synchrotron DESY, Hamburg, Germany. [2]The Hamburg Centre for Ultrafast Imaging, Hamburg, Germany. [3]Department of Physics, University of Hamburg, Hamburg, Germany. [4]Centre for Structural Systems Biology CSSB, Hamburg, Germany. [5]European Molecular Biology Laboratory Hamburg, Hamburg, Germany. [6]Centre for Structural Systems Biology CSSB, Deutsches Elektronen-Synchrotron DESY, Leibniz Institute of Virology (LIV), University of Lübeck, Hamburg, Germany. [7]Institute of Chemistry and Metabolomics, University of Lübeck, Lübeck, Germany. ✉e-mail: thomas.lane@desy.de

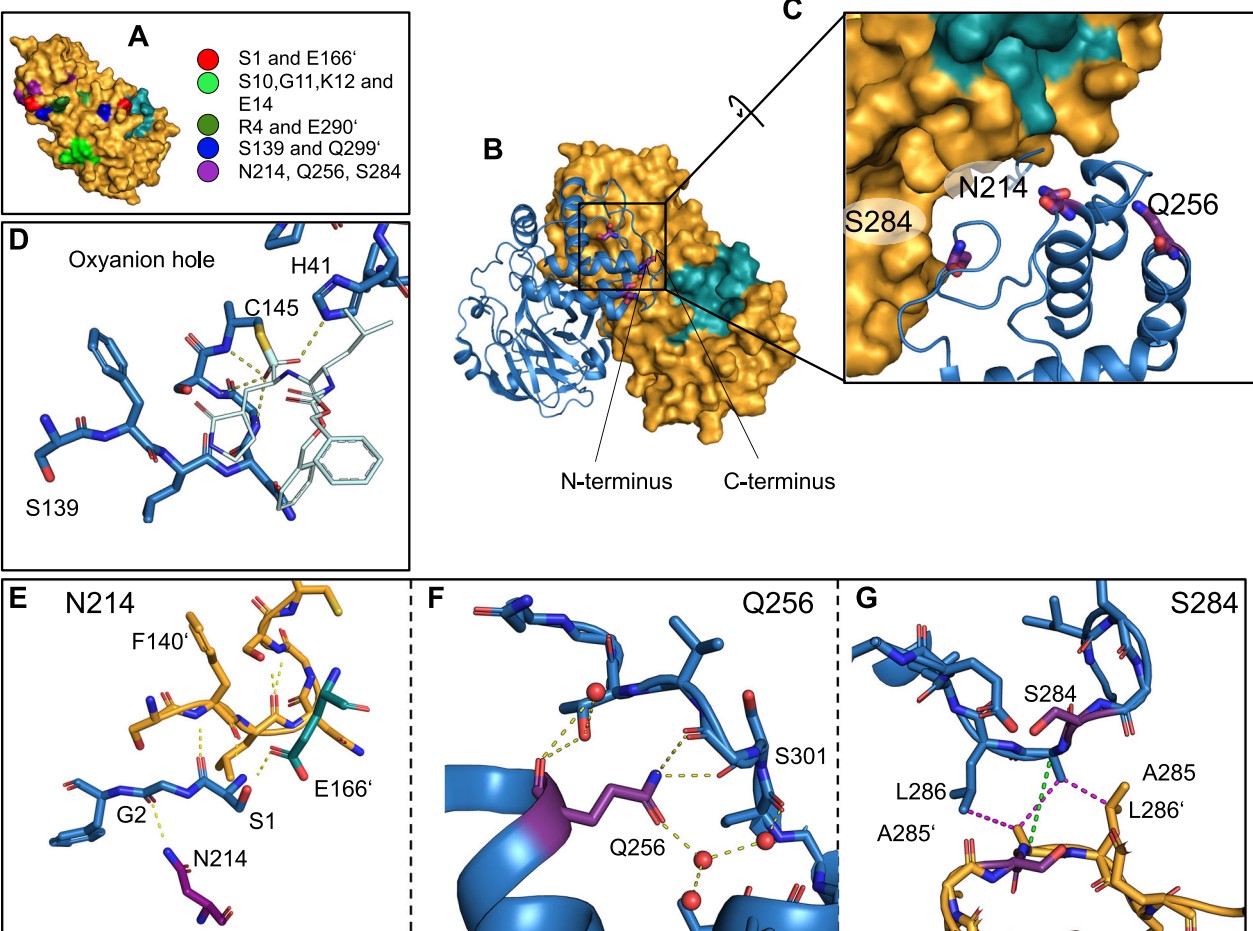

**Fig. 1 | Structure of the M^pro homodimer.** Dimerization is obligate for M^pro enzyme turnover. **A** Through mutagenesis studies, several residues on the dimer interface have been identified as essential for the formation or function of the dimer assembly. These sites on the dimer interface (green) are generally distal from the active site. As part of this work, we identified three residues (purple) that are predicted to have strong structural correlation with the active site (teal). **B** The dimer assembly. **C** Highlighted are the three residues we mutated in this work (purple). N214 and S284 are located at the dimer interface, while the final residue, Q256, is solvent exposed. **D** The active site oxyanion hole (G143, S144, and C145) is formed by backbone amide groups along the oxyanion loop (S139–C145). The ability of this structure to stabilize a tetrahedral transition state is highlighted in structures of transition state analogs, such as the covalent inhibitor GC376, shown here forming a hemithioacetal linkage (PDB 7D1M). The hydroxyl portion of the hemithioacetal is disordered and sits in two positions that approximate the tetrahedral transition state geometry. **E** N214 (purple) forms a hydrogen bonding network with the N-terminus, which in turn interacts strongly with a loop (S139 to C145) that contains the catalytically essential oxyanion hole in the active site and harbors the central catalytic thiol. **F** Q256 (purple) is largely solvent exposed, but is adjacent to the C-terminus, which is stabilized by hydrogen bonds between the Q256 sidechain and S301 backbone. Finally, **G** S284 (purple) forms part of a hydrophobic zipper (included interactions are shown in magenta) that bridges the dimerization domains of both dimer-forming protomers. The distance of 4.8 Å between the C_α atoms of A285 and A285' is highlighted in green. In all panels, hydrogen bonds are shown in yellow (2.2 to 3.5 Å). All panels show PDB 7BB2[67], except (**D**), which shows 7D1M[85].

identified four major sites as important for dimerization, shown in Fig. 1A. First, a hydrogen bond between residues S1 and E166', where a prime indicates the dimeric protomer, is essential to stabilize the active site pocket. Mutation of either residue modulates the activity of the enzyme[22]. Second, S10, G11, K12, and E14 form hydrogen bonding interactions with the same residues, S10'-E14', on the dimeric protomer. Mutation of any of these to alanine degrades activity[22]. Third, R4 and E290' form a salt bridge across the dimer interface; mutants of either residue are less active[22,23]. Fourth, S139 and Q299' form a key dimer-bridging hydrogen bond, and mutation of Q299 to alanine produces inactive dimers[22]. Finally, through specific point mutation or deletion of the C-terminal dimerization domain, monomeric variants of M^pro have been generated that show a collapsed oxyanion hole with a 3_10 helix spanning S139–N142 (Supplementary Fig. 13A). This conformation has been proposed to be the predominant inactive conformation in the wild-type monomeric state[20,24]. Combined, all these mutational studies pinpoint key interactions on the dimer interface but have not been able to reveal the global picture of atomic correlations of the M^pro native state.

Here, we employ high-throughput cryo-crystallography to address this gap and reveal the correlated fluctuations in M^pro's native state. Cryo-crystallography has not traditionally been thought of as an experiment one performs to gain insight into the structural fluctuations of proteins. The structures used to model protein crystal diffraction are predominantly single conformers, augmented with B-factors and a few alternative sidechain conformations that capture the limited structural heterogeneity within the crystal system. Indeed, strong structural coherence between the proteins that form the lattice is a necessary requirement for high-resolution diffraction. Therefore, we expect protein crystals to yield a narrow distribution of structures around a mean, with this mean reported as "the structure".

By contrast, large sets of crystal structures can produce diverse ensembles enabling the study of protein flexibility[25]. Employing this approach on M^pro itself, an analysis of multiple crystal structures deposited in the PDB allowed Kidera et al. to interrogate the distribution of structures M^pro could adopt[26]. Inspecting structures that included bound ligands and SARS-CoV-1 sequences, they highlighted the importance of the hydrogen bond chain formed between N214, S1, G2, and F140', which crosses the

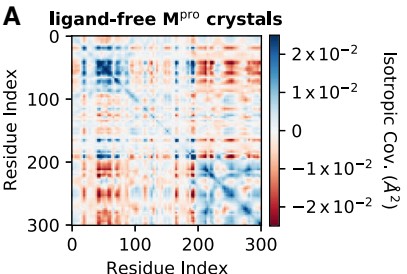
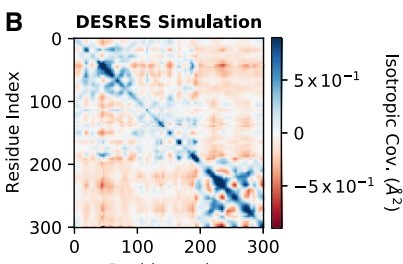
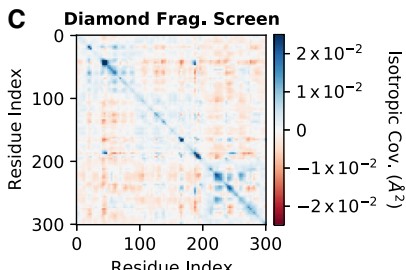

**Fig. 2 | Set of crystal structures is consistent with correlated fluctuations predicted by MD simulation.** The isotropic covariance all of C$_\alpha$ positions in M$^{pro}$ structures contained in three different sets of structures: **A** 1146 ligand-free, automatically refined M$^{pro}$ structures obtained as part of a ligand screening campaign[31] and analyzed in this work, **B** a 100 μs molecular dynamics simulation performed by D. E. Shaw Research[32], **C** 95 ligand-bound, hand-refined M$^{pro}$ structures released by Diamond Light Source[35]. Blue coloration indicates positive covariance; red, anti-covariance; white, no measurable covariance. The covariances, reported in Å$^2$, are scaled as if they were crystallographic B-factors (see "Methods and Materials"). Note that the color scale for the simulation in (**B**) is approximately 4-fold larger than for the experimental datasets, as the covariance magnitudes for the simulations are considerably larger.

dimer interface and connects to a loop containing the active C145. This work expresses a key idea: that by interrogating a large number of distinct cryogenic crystal structures, it may be possible to extract meaningful information about global atomic correlations in M$^{pro}$ and other protein systems, an idea that has been used to study allostery in epidermal growth factor receptor (EGFR)[27], RAS[28], and more recently the protein tyrosine phosphatases (PTPs)[29]. Further supporting this idea, Shen, Robertson, and Bax found that a set of 350 M$^{pro}$ crystal structures from the PDB outperformed other multiconformer models in predicting NMR residual dipolar couplings[30], suggesting that this set of crystal structures contains relevant information about the enzyme in the solution phase.

Here, we show that statistically significant samples of crystal structures can be collected proactively, in a single experiment, then used to study protein flexibility and gain insight into function. This work builds from our recently reported crystallographic ligand screening campaign against M$^{pro}$. In addition to crystal structures bound to novel ligands, our screening effort generated many thousands of ligand-free structures in our search for binders, orders of magnitude more than typically studied. As we refined structural models to each dataset in our search for ligands, we observed that the resulting structures exhibited a large degree of structural heterogeneity and hypothesized that this distribution of structures might yield new information about the function of M$^{pro}$. Intriguingly, our crystal dataset contained a distribution of structures consistent with those from molecular dynamics (MD) simulation. Going beyond the state of the art, we used our structures to infer a model of the protein's flexibility, which in turn revealed an allosteric network connecting the enzyme's dimerization domain to its active site. We then attempted to falsify this model, characterizing mutants predicted to knockout the allosteric pathway and seeking explanations for the physical origins of the structural diversity observed. Our work extends established analyses using sets of crystal structures by inferring a model of the covariance of atomic displacements, then using that model to make specific and falsifiable predictions about protein function.

## Results
### Large-scale ligand screening results in a distribution of structures

In the spring of 2020, with the objective of discovering novel M$^{pro}$ inhibitors, we partnered with other investigators and performed a large-scale crystallographic screening campaign[31]. M$^{pro}$ was co-crystallized in the presence of compounds from two repurposing libraries comprising 5953 unique molecules, resulting in the discovery of 37 binders with interpretable electron density[31]. During this screening effort, we determined and automatically refined structures for 6976 individual crystals of M$^{pro}$, the vast majority of which showed no bound ligand. These structures exhibited surprising structural variability, which we decided to investigate further.

To remove potential complicating factors due to ligand binding, we discarded any structure with observed ligand-bound density. We further discarded any other dataset that originated from crystals grown in the presence of a ligand known to bind, either from crystallographic or biophysical characterization, whether or not there was observable ligand density in the crystal structure. Further, to ensure quality, datasets were further down-selected based on resolution (2.2 Å or better) and model-data fit ($R_{free}$ at least 0.25 or lower), producing a final set of 1146 structures for subsequent analysis (Supplementary Figs. 1 and 2).

### Covariance analysis of the set of crystal structures

Given its structural diversity, we wondered if the set of crystal structures might provide information about the protein's solution-phase native state behavior. To test this hypothesis and analyze the conformational heterogeneity present in our crystal structures, we computed the pairwise covariances between each C$_\alpha$ atom in the protein (Fig. 2, "Materials and Methods"). In our crystal system, the two protomers that form the dimer complex are crystallographically symmetric; the covariance between two atoms in the same protomer is identical to the covariance across protomers, subject to a rotation. Therefore, for simplicity, we report a single monomer in most analyses. Other measures of structural correlation were investigated, which either are consistent with our conclusions (Supplementary Fig. 3) or did not converge to meaningful results given the quantity and quality of structures available (Supplementary Fig. 4), and a clustering analysis failed to identify clearly distinguished groupings as the data form a largely continuous manifold (Supplementary Fig. 5).

We compared the C$_\alpha$ covariance from our set of crystal structures to a 100 μs MD simulation of M$^{pro}$ in solution conducted by D. E. Shaw Research[32], as well as two shorter simulations conducted by independent groups (Supplementary Fig. 6)[33,34]. The C$_\alpha$ covariances computed from both the set of crystals and MD simulation show notable qualitative agreement (Fig. 2). To quantify this similarity, we computed the Forbenius norm between different covariance matrices and found that the covariance matrix inferred from crystals is as similar to the MD simulations as the MD simulations are to one another (Supplementary Fig. 6). While the MD simulation samples a significantly wider conformational space (Supplementary Fig. 2), we cannot exclude effects due to the crystal lattice, and the covariance comparison is fundamentally qualitative, the similarity of the covariance around the native conformation supports the hypothesis that structural variability in the crystal set reflects an important subset of the protein's solution-phase conformational ensemble.

In contrast, ensemble refinements, where multiple structures are used to fit a single crystallographic dataset, failed to match the covariance matrices of the MD simulation and set of crystal structures (Supplementary Table 1 and Supplementary Figs. 7, 8). Later, we investigated why the protein conformations in distinct crystals might differ more than the set of

## Table 1 | Crystallographic statistics for the N214A, Q256A, and S284A mutants

| Mpro <br> PDB ID | N214A <br> 9GI6 | Q256A <br> 9GHN | S284A <br> 9GHO |
|---|---|---|---|
| Data collection | | | |
| Source | PETRA-III | PETRA-III | PETRA-III |
| Temperature | 100 K | 100 K | 100 K |
| Space group | $P2_12_12_1$ | $P2_12_12_1$ | C2 |
| $a, b, c$ (Å) | 67.81, 100.57, 102.18 | 78.54, 88.44, 100.77 | 113.20, 53.27, 44.84 |
| $\alpha, \beta, \gamma$ (°) | 90, 90, 90 | 90, 90, 90 | 90, 102.4, 90 |
| Resolution cut | Anisotropic | Isotropic | Isotropic |
| Resolution (Å)[a] | 55.28–2.51 55.28–2.58 55.28–1.83 (2.01–1.83) | 66.47–1.40 (1.45–1.40) | 55.28–2.00 (2.07–2.00) |
| Wilson B (Å²)[a] | 45.29, 77.34, 58.51 | 19.09 | 39.02 |
| $R_{merge}$ | 0.080 (0.400) | 0.049 (1.505) | 0.110 (0.825) |
| $R_{meas}$ | 0.086 (0.438) | 0.053 (1.649) | 0.127 (0.963) |
| $R_{pim}$ | 0.031 (0.221) | 0.020 (0.6602) | 0.0627 (0.487) |
| $I / \sigma I$ | 12.1 (1.5) | 19.05 (1.05) | 5.72 (0.54) |
| $CC_{1/2}$ | 0.997 (0.943) | 0.999 (0.584) | 0.995 (0.421) |
| Total reflections | 320 497 (15 980) | 992 672 (78754) | 67 221 (6465) |
| Unique reflections | 42 912 (2156) | 137736 (13221) | 17512 (1734) |
| Multiplicity | 7.5 (7.4) | 7.2 (6.0) | 3.8 (3.7) |
| Completeness (%) | 94.8 (64.5) | 99.42 (95.29) | 95.55 (85.99) |
| Refinement | | | |
| $R_{work}$ | 0.174 (0.217) | 0.193 (0.3761) | 0.188 (0.321) |
| $R_{free}$ | 0.207 (0.304) | 0.218 (0.4038) | 0.234 (0.318) |
| No. atoms | 5229 | 5537 | 2458 |
| Protein | 4755 | 4883 | 2366 |
| Ligand/ion | 164 b | 16 | 9 |
| Water | 310 | 638 | 83 |
| B-factors | 36.26 | 25.53 | 41.77 |
| Protein (Å²) | 35.50 | 24.50 | 41.77 |
| Ligand/ion (Å²) | 52.28 | 43.74 | 51.01 |
| Water (Å²) | 39.41 | 33.03 | 40.70 |
| r.m.s. deviations | | | |
| Bond lengths (Å) | 0.007 | 0.005 | 0.007 |
| Bond angles (°) | 0.84 | 0.83 | 1.03 |

[a]*staraniso* produces an elliptical diffraction limit (1); for the N214A dataset, we report the resolution cutoff and Wilson B factors along the three principal elliptical axes; statistics take this elliptical truncation into account.

[b]22 PEG chains (7 atoms/chain) were modeled into the density observed in the N214A dataset. See "Materials and Methods" for crystallization conditions of each variant.
Statistics for the highest-resolution spherical shell are shown in parentheses.

conformations within a single crystal (see "Structural diversity is a function of crystal lattice perturbation", below).

To evaluate how robust this finding was, we compared our set of crystals to a publicly released set of 95 Mpro structures from Diamond Light Source[35]. These structures were refined by hand and bound to fragments, in contrast to our ligand-free automatically refined models. The Diamond data, containing an order of magnitude fewer structures, has a considerably weaker signal-to-noise ratio. Nonetheless, at the level of $C_\alpha$ covariance, the two sets of crystals show qualitative agreement (Fig. 2).

### Covariance pinpoints covariance hotspots on the dimerization domain

Encouraged by the agreement between our set of crystals and MD simulations, we pursued the idea that the crystal structures might be used to infer structural fluctuations implicated in biological function. Specifically, we investigated how dimerization regulates catalysis by Mpro.

To do this, we adopted a simple approach and measured the average $C_\alpha$ covariance of all residues with the active site (Fig. 3 and Supplementary Fig. 8). We attempted to capture information regarding the positions of sidechains as well, for instance by measuring the mutual information of $\chi_1$ angles ("Materials and Methods", Supplementary Fig. 4), but could not obtain converged, reliable estimates with the data we had in hand. Proceeding with the backbone analysis, three motifs of highly covarying substructures emerged. The first were mobile loops and surface-exposed amino acids directly proximal or contiguous with the active site, many of which bridge the dimer interface, most notably the N-terminus. More surprisingly, the second group contained a set of residues that form crystal contacts. The final category represented three regions of the dimerization domain, with high covariance peaking at N214, Q256, and S284 (Fig. 3), highlighting that these locations are structurally coupled to the active site.

### Mutational studies test predicted covariance hotspots

We speculated that these residues might play an especially important role in the structural transitions that activate Mpro upon dimerization. Figures 1B and 1C show the dimer structure with amino acids N214, Q256, and S284 highlighted in purple. Lacking a quantitative theory or simulation that could map our covariance model to biophysical (dimerization affinity) and biochemical (catalytic rate) properties, we formulated a qualitative hypothesis. We predicted we could perturb the structure of these sites (N214, Q256, and S284) through mutation. If, as our model predicts, these sites were in fact correlated with the active site, such mutations would in turn affect the active site structure and thereby the enzyme's catalytic properties, almost certainly degrading activity. Such a reduction in activity could be caused by two non-exclusive mechanisms: first, by weakening the dimer affinity, thereby reducing the fraction of active dimers, or second, through a degradation of the catalytic capabilities of the dimers themselves.

Due to the strong correlation between N214/Q256/S284 and the active site, we predicted that both effects would be observed. Therefore, we refer to these specific sites on the dimer interface, where structural change influences the dimer activity (and not just the concentration of dimers), as *putative covariance hotspots*.

We set out to test this hypothesis and chose to do so by making alanine substitutions at the N214, Q256, and S284 hotspot locations, then characterizing these mutants through a series of biophysical, biochemical and crystallographic experiments. Because alanine substitution is a specific perturbation that may or may not significantly disrupt the structure at the site of the mutation, if these alanine mutants disrupted the enzymatic function of Mpro, we elected to call them *alanine-confirmed covariance hotspots*. We note that if a specific alanine substitution does not perturb the enzyme structure significantly, we cannot definitively conclude the site is not a hotspot.

First, we determined crystal structures for each of the mutants (Fig. 4 and Table 1). These structures allow us to assess if the mutation in fact perturbs the mean conformation of Mpro, both at the site of mutation and in the active site. Further, our structures provide insight into whether specific regions of the protein become ordered or disordered upon mutation.

Second, we assessed the biochemical and biophysical consequences of our mutations. We interrogated monomer/dimer equilibrium with analytical size exclusion chromatography (SEC, Fig. 5) and native mass spectrometry (MS, Fig. 5, Supplementary Table 3, and Supplementary Fig. 9), allowing us to establish the dimer affinities of each Mpro variant (Table 2 and Fig. 5). Further, we independently assessed ligand binding affinity with isothermal titration calorimetry (ITC) and nano differential scanning fluorimetry (nDSF, Table 2, Fig. 5, Supplementary Figs. 10 and 11). To avoid

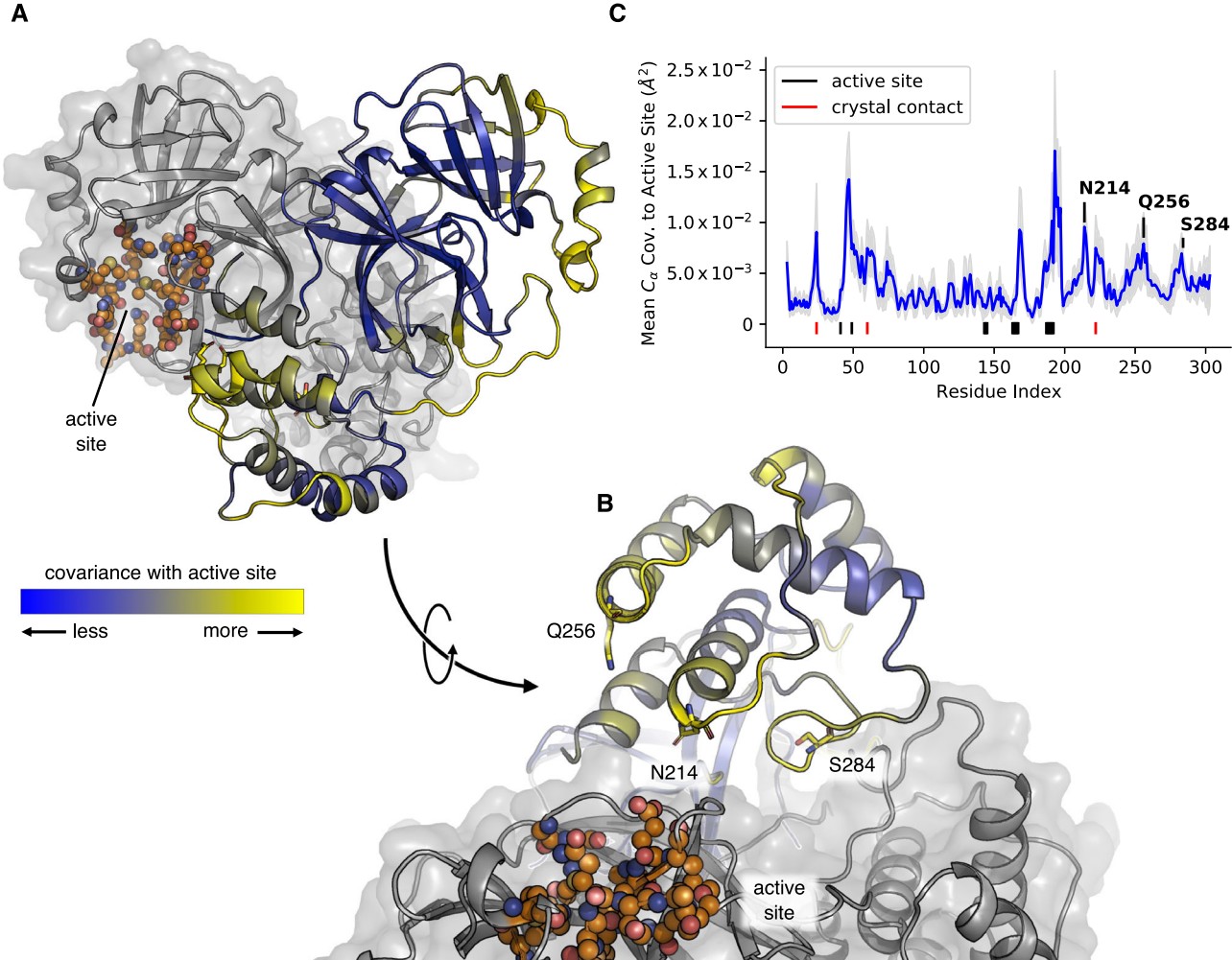

**Fig. 3 | Covariance model predicts how the dimer interface affects the active site structure. A, B** Two views of wild-type M$^{pro}$, rotated by 90° with respect to one another, showing three predicted covariance hotspots (N214, Q256, S284; sticks), the active site (orange spheres) and the dimeric structure of M$^{pro}$ (one protomer in blue/yellow cartoon, the other in grey cartoon and surface). The average C$_\alpha$ covariance to the active site residues is indicated by the cartoon color of one protomer, with blue indicating regions of less covariance, and yellow higher covariance. **C** The same average C$_\alpha$ covariance as a function of residue index (blue trace, per-residue), highlighting active site residues (black bars below blue trace), crystal contacts (red bars), and the three identified hotspots on the dimerization domain. An estimate of the statistical error was determined using a 1000-fold bootstrap with replacement (see "Methods and Materials"), the shaded grey region indicates three standard deviations.

complications due to substrate turnover in these experiments, we used calpeptin, a reversible covalent M$^{pro}$ inhibitor, as a probe compound because it binds in a substrate-like mode and is well-characterized[36].

Finally, to assess the effect of our mutations on the catalytic activity itself, we used a fluorescent peptide cleavage assay and characterized turnover kinetics (Fig. 5 and Table 2, *Materials and Methods*). Since previous studies found that both wild-type and mutant M$^{pro}$ follow Michaelis–Menten kinetics[18,37,38], we expected our mutants would as well. We were agnostic as to whether the predicted reduction in activity would be caused by a reduction in $k_{cat}$, $k_{cat}/K_M$, or both, as structural perturbation of an active site could reasonably affect rates of binding, dissociation, catalysis, or any combination thereof.

Kinetics schemes that attempted to model both the dimerization and turnover in a holistic manner (Supplementary Fig. 12 and Supplementary Table 4), could not account for the kinetics of one of our mutants. We found, however, that at each enzyme concentration, a Michaelis-Menten model describes observed initial turnover velocities. We therefore found it most productive to simply analyze $k_{cat}$ and $k_{cat}/K_M$ as a function of enzyme concentration. At enzyme concentrations well above the monomer/dimer $K_D$, the Michaelis-Menten parameters characterize the dimer form; at intermediate concentrations, they characterize an average of the monomers and dimers present. Our biophysical experiments enabled us to determine the monomer/dimer $K_D$ and therefore make direct comparisons between our different M$^{pro}$ variants.

**Variant N214A.** Of the three variants we studied, the mutation of N214 to alanine induces the most significant changes in enzyme structure and function. The N214 sidechain interacts in a hydrogen bond network that bridges the dimer interface, involving the N-terminus of the same protomer and a specific loop containing the oxyanion hole of the dimeric protomer. Our results revealed the turnover rate of the N214A mutant is significantly suppressed compared to wild type, supporting previous work in which an alanine scan of SARS-CoV-1 M$^{pro}$[39-41] found N214A was catalytically compromised. Our experiments reveal the structural origins and biophysical consequences of this mutation on enzyme function.

Both analytical SEC and native MS show that the N214A mutation significantly lowers the dimer affinity (Fig. 5, Table 2, Supplementary Table 3 and Supplementary Fig. 9). Due to an increased dissociation rate, the variant does not elute during SEC experiments as stable monomer and dimer species. Consequentially, native MS provides the most reliable estimate of the N214A monomer/dimer $K_D$ of 5.6 ± 0.9 μM, more than 10 times larger (weaker) than wild type (0.5 ± 0.2 μM).

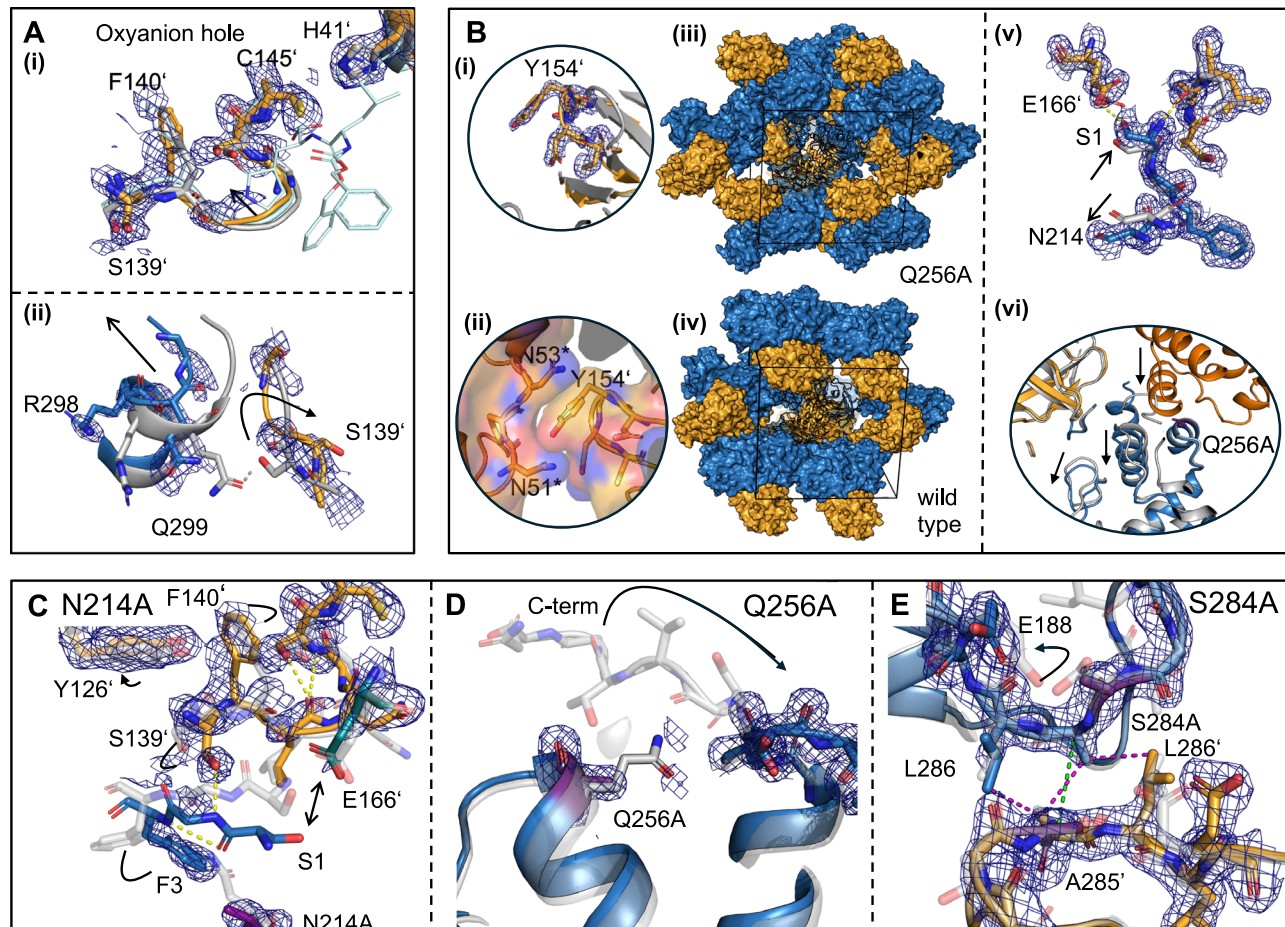

**Fig. 4 | Structures of the N214A, Q256A, and S284A mutants.** Mutant structures are shown in blue (chain A) and orange (chain B) with a wild-type structure (light grey) as a reference. Two wild-type structures were employed in order to match the space group and lattice parameters between the reference and mutant structures as closely as possible, specifically PDB: 7BB2 for the N214A (PDB: 9GI6) and Q256A (PDB: 9GHN) mutants ($P2_12_12_1$) and PDB: 7AR6 for the S284A (PDB: 9GHO) mutant (C2). All densities shown are $2mF_o$-$DF_c$ maps at 1 RMSD. **A** Notable changes in the N214A mutant crystal structure (PDB: 9GI6). (i) Removal of the N214 sidechain disrupts a hydrogen bonding network (which is shown in **C**), displacing F140' and S139', which form part of the oxyanion loop in the active site of the dimeric protomer. Shown in cyan is the structure of the bound inhibitor (GC376, PDB: 7D1M), also shown in Fig. 1 as a reference. (ii) This displacement further disrupts a hydrogen bonding interaction between S139' and Q299 (grey). Q299 is located in an α-helix close to the C-terminus, which is displaced from the dimer interface. **B** Notable changes in the Q256A mutant structure (PDB: 9GHN). Upon mutation, two new crystal contacts form. (i) The first contact occurs due to the flip of a β-hairpin (D153–C156), which inserts the phenol sidechain of Y154 into a pocket formed by N51*–N53* of an adjacent dimer, an interaction stabilized by hydrogen bonds between these residues (asterisks denote distinct symmetry-related units in the crystal). (ii) In the second contact, the α-helix comprising amino acids L227** to

Y237** from a second neighboring dimer fills a hydrophobic pocket generated by the removal of the Q256 sidechain. (iii) This results in a new crystal lattice, distinct from (iv) the wild type. Moreover, (v) the interaction between S1 and E166' is impacted by a displacement of the α-helix containing N214. This movement is accompanied by a small shift in the position of S1, increasing its distance from E166' and weakening their interaction. (vi) Consequently, the C-terminus is displaced. **C–E** Details of the variant structures at the site of mutation, with the mutated residues highlighted in purple. **C** Changes in the hydrogen bond network in the N214A mutant. The loss of the carboxamide sidechain at N214 precludes a stabilizing interaction with G2 in the N-terminus of the same protomer. This results in the flip of F3, F140', and S139', the disruption of hydrogen bonds between S1 and the backbone of P140' and S1 and E166', and the formation of new hydrogen bonds between S1 and F3 and G2 and S139'. The loss of the hydrogen bond network leads to shifts in the atomic positions of C143, G144, and C145. The oxyanion hole on the dimeric protomer shifts, even though the hydrogen bond between S144 and L141 still exists (Supplementary Fig. 13). **E** The hydrophobic zipper at the dimer interface that contains S284 is nearly unchanged upon mutation of this residue to alanine. Only an elongation of the inter-protomer distance from 4.8 to 5.5 Å (A285–A285' $C_\alpha$, green) was observed (PDB: 9GHO, blue/PDB: 7AR6, grey).

This weakening of the dimer affinity is expected to degrade the enzyme's observed turnover. Our biochemical experiments, however, show that the reduction in N214A's activity cannot be fully accounted for by the reduction in dimer concentration. The remaining N214A dimers are significantly less active than wild type (Fig. 5). We compared $k_{cat}$ and $k_{cat}/K_M$ as a function of enzyme concentration (Fig. 5D), revealing how these parameters, which represent an average of the dimer and monomer properties, change as a function of monomer and dimer concentrations. Using monomer/dimer $K_D$s determined from native MS, we can estimate the fraction of dimeric enzyme specifically, enabling a direct comparison (Fig. 5E, see Supplementary Fig. 12 for fits). We find that both $k_{cat}$ and $k_{cat}/$

$K_M$ are suppressed when compared to wild type. Concretely, at the highest enzyme concentrations measured, 8 µM and 64 µM for wild type and N214, respectively, >75% of either variant is expected to be dimeric. Comparing these concentrations, the wild-type $k_{cat}$ is 8x and $k_{cat}/K_M$ is 40x greater than measured for N214A. This implies that the reduction in activity observed in N214A is mostly due to a reduction in the rate of chemical catalysis, and that compromised ligand binding is a secondary effect. Supporting this, ITC and nDSF experiments show N214A binds a substrate analog, calpeptin, as strongly (ligand binding $K_D$ 2.0 ± 0.4 µM) as the wild-type enzyme (2.4 ± 0.8 µM, Table 2, Supplementary Table 2, Supplementary Fig. 10, and Supplementary Fig. 11).

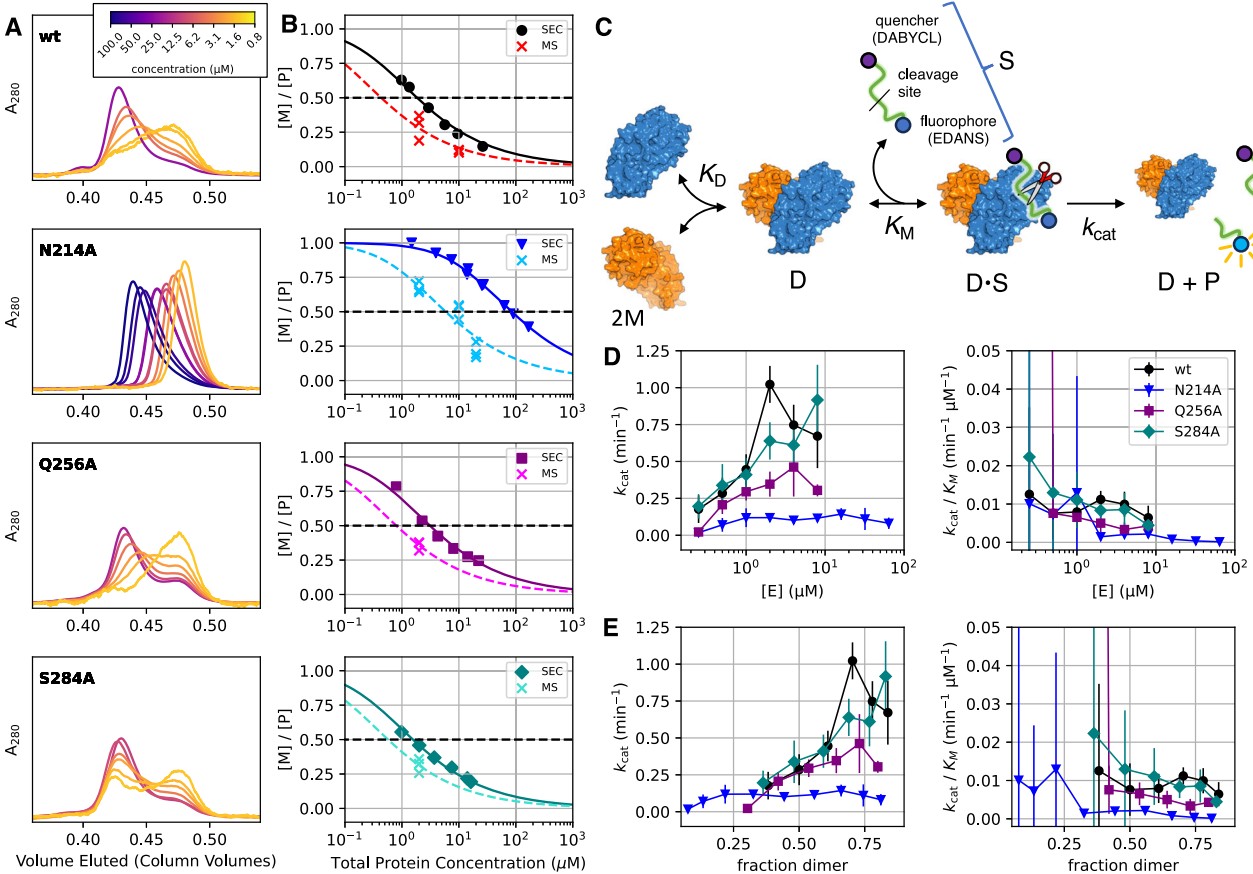

**Fig. 5 | Analytical SEC, native MS, and peptide cleavage assay reveal the functional consequences of mutation at predicted covariance hotspots. A** Analytical SEC as a function of protein concentration. The wild type, Q256A and S284A variants show two distinct elution peaks corresponding to dimer and monomer species, demonstrating these species are stable on the timescale of the experiment (~15 min). In contrast, the N214A variant elutes as a single, skewed peak at all concentrations, consistent with rapid exchange on the timescale of the experiment. **B** Two-state model explains monomer/dimer data. The SEC data (dark, filled symbols) show two-state dissociation behavior for all variants, yielding dimer affinities of approximately 2 µM for the wild type, Q256A and S284A variant. In contrast, the dimer affinity of N214A is substantially reduced (Table 1). Native mass spectrometry measurements (crosses, light colors) have a limited concentration range, but do not suffer from dilution over the course of the experiment, as with SEC. Therefore, $K_D$s determined by native MS data provide a corroborative monomer/dimer $K_D$ (Table 1, Supplementary Table 4 and Supplementary Fig. 5), which in the case of rapidly exchanging N214A is significantly more reliable. Lines, both solid and dashed, show the predicted fraction of monomer species for the corresponding $K_D$ values reported in Table 2. $K_D$s determined from SEC data were fit using all data points shown, while $K_D$s from native MS data were determined from triplicate measurements at 2 µM, to minimize concentration-dependent systematic error ("Materials and Methods", for all native MS $K_D$s see Supplementary Table 3). **C** Characterization of the enzyme kinetics via a peptide cleavage assay shows that **D** the Q256A and N214A variants have decreased activity compared to wild type and S284A variants. Michaelis–Menten models were used to determine observed $k_{cat}$ and $k_{cat}/K_M$ parameters as a function of enzyme concentration, where the fit Michaelis–Menten reflects an average of the present dimer and monomer species. **E** Using $K_D$s determined from native MS and plotting these parameters as a function of the fraction of dimer concentration reveals that the reduction in Q256A's activity can be wholly attributed to a reduction in dimer affinity, while dimers of N214A are systematically slower catalysts, even at enzyme concentrations well above the monomer/dimer $K_D$. See Supplementary Fig. 12 for alternative kinetics models. Error bars represent 95% confidence intervals for fit parameters. Note that at low enzyme concentrations, the turnover velocities are very small, and the corresponding errors for $k_{cat}$ and $k_{cat}/K_M$ are large.

Our structure of N214A provides insight into why the loss of the N214 sidechain disrupts dimerization and catalysis so dramatically. M$^{pro}$ employs an oxyanion hole (G143 to C145), embedded in a loop (S139 to C145) which we refer to as the oxyanion loop, to stabilize the reaction transition state highlighted in structures of transition state analogs (Fig. 1). In the wild-type system, the sidechain of N214 participates in a hydrogen-bonding network mediated by the N-terminus that stabilizes this oxyanion loop (Fig. 1E). Recent MD simulations pinpointed the N-terminus as a key mediator of the cooperativity between the two dimeric active sites[42]. In our N214A structure, in which the protein forms a native dimer, this network is disrupted and the oxyanion loop is only partially ordered (Figs. 4A(i), 5C and Supplementary Fig. 13). Since ITC and nDSF measurements show N214A is a competent substrate binder, we propose that removal of the N214 sidechain reduces the population of conformations capable of stabilizing the reaction transition state, in turn compromising enzyme activity.

Protomer B of our structure shows this disruption most prominently. Specifically, lacking the stabilizing interactions provided by N214, the N-terminus shifts, breaking hydrogen bonds with E166' that hold the N-terminus in place (Fig. 4C). This disrupts a series of hydrogen bonds between the oxyanion loop and N-terminus that normally stabilize the catalytic conformation (Fig. 4A, C). The loss of the carboxamide sidechain at N214 precludes a hydrogen bond with G2 in the N-terminus of the same protomer. This results in a flip of the amino acids F3, S139', and F140', the disruption of additional hydrogen bonds between S1 and the backbone of P140' and S1 and E166', the formation of new hydrogen bonds between S1 and F3 and G2 and S139'. The loss of the hydrogen bond network leads to shifts in the atomic positions of C143, G144, and C145 (SI Fig. 13A). The mean position of the oxyanion hole on the dimeric protomer shifts, even though the hydrogen bond between S144' and L141' persists (Supplementary Fig. 13). Additionally, a hydrogen bond between Q299 and S139' (Fig. 4A(ii)) is exchanged for one between S139' and N-terminal residue G2,

**Table 2 | Monomer/dimer and ligand binding affinities of M^pro variants**

| variant | Analytical SEC Mono./dimer $K_D$ (µM) | Native MS Exchange | ITC Mono./dimer $K_D$ (µM) | Ligand $K_D$ (µM) |
|---|---|---|---|---|
| wild type | 1.84 ± 0.15 | slow | 0.5 ± 0.2 | 2.4 ± 0.8 |
| N214A | <90 ± 4 | fast | 5.6 ± 0.9 | 2.0 ± 0.4 |
| Q256A | 3.1 ± 0.3 | slow | 0.80 ± 0.11 | 3.4 ± 1.0 |
| S284A | 1.55 ± 0.08 | slow | 0.56 ± 0.11 | 1.6 ± 0.4 |

Analytical SEC and native MS data were fit to a two-state binding model to estimate a monomer/dimer $K_D$ (Materials and Methods, Fig. 5). Since N214A did not elute as stable monomer and dimer species during analytical SEC experiments, a model assuming fast exchange was fit ("Materials and Methods") and accordingly, the corresponding $K_D$ for this variant should be considered an upper bound. $K_D$s determined by native MS are systematically lower than those determined by SEC, which we attribute to dilution of the protein in SEC experiments. Further, the affinity for substrate-like compounds was evaluated by measuring a ligand binding $K_D$ for the inhibitor and substrate analog calpeptin. ITC experiments were conducted above the monomer/dimer $K_D$ for all variants, approximately 20 µM M^pro (Supplementary Table 2). Further, the consistent ligand binding affinity for wild type and N214A is supported by orthogonal nDSF measurements (Supplementary Fig. 11). All reported uncertainties are standard errors.

resulting in a disordered C-terminus (residues 301–306), which normally participates in the dimer interface (Fig. 4C). Without these stabilizing interactions that bridge the two protomers, the oxyanion loop collapses, adopting a partially disordered conformation in which the C143, G144, and C145 amides are not oriented to support catalytic activity. In contrast to the monomeric structure Mpro1-199 construct (PDB: 7UJ9), in our N214A structure, the loop formed by residues S139 to L141 does not form a $3_{10}$ helix. Instead, we detected a loss in observable density and a shift in the mean position of residues L141 to G143, potentially reflecting an intermediate state between the monomer construct and wild type. (Fig. 4A(i) and Supplementary Fig. 13). Dimerization and catalytic turnover are compromised as a result, confirming N214 is an alanine-confirmed hotspot and plays a pivotal role in the allosteric network that connects dimerization to function in M^pro.

**Variant Q256A.** The second putative covariance hotspot we identified, Q256, forms part of M^pro's dimerization domain but sits far from the dimer interface. In our wild-type structures the Q256 sidechain is solvent-exposed. No key interactions of this residue with any other are apparent in our structures, nor have been reported to date in other literature.

In contrast to the N214A mutant, native MS and SEC measurements show that Q256A exhibits moderately reduced dimer affinity about two times weaker than wild type (Fig. 5, Table 2, Supplementary Fig. 9, and Supplementary Table 3). Biochemically, substituting Q256 for alanine results in a modest decrease in $k_{cat}$ and $k_{cat}/K_M$ vs. wild type (Fig. 5 and Supplementary Fig. 12). At 8 µM of both variants, the wild-type $k_{cat}$ and $k_{cat}/K_M$ are twice as large as for Q256A. The majority of this decrease in activity can be accounted for by the reduction in dimer concentration due to the weaker monomer/dimer $K_D$ (Table 2 and Fig. 5E). Interestingly, ITC measurements show Q256A binds calpeptin inhibitor about twice as weakly as the other variants studied. Therefore, as compared to wild type, Q256A demonstrates a weaker affinity for substrate analog and a weaker dimer affinity, which in turn causes a reduction in enzymatic activity.

This weakening of the dimer complex is surprising given that, in wild-type structures, Q256 is spatially removed from the dimer interface. Seeking a structural explanation, we crystallized the Q256A mutant under the same conditions as the wild-type enzyme. Interestingly, this produced crystals in a distinct lattice not found in any of the more than 1600 M^pro structures deposited in the PDB at the time this manuscript was prepared (Table 1). Removal of the polar Q256 sidechain creates a hydrophobic pocket that is filled by a helix from a neighboring dimer assembly, consisting of residues L227*–Y237* (where we use * and later ** to indicate distinct biological units in the crystal lattice). This helix-in-pocket packing forms a prominent

crystal contact that supports this lattice and displaces the C-terminus, which sits at the crystal interface (Fig. 4D). A second new crystal contact is formed when a β-hairpin (D153-C156) flips so that the phenol sidechain of Y154 can insert into a pocket formed by N51–N53 on an adjacent dimer (Fig. 4B(i), (ii)). When compared to wild-type structures (e.g., PDB: 7AR6 and PDB: 7BB2), the dimerization domain—including the helix that contains N214—reorients substantially with respect to the rest of the protein (Fig. 4B(v), (vi)).

The site of mutation in our Q256A structure is directly involved in a new crystal contact, and there is no wild-type structure with the same contacts. Therefore, while the dimer interface is substantially perturbed in our Q256A structure, we cannot say if this is due to the novel crystal lattice or an effect due to the mutation that would be retained for the protein in solution. Attempts to obtain crystals with alternative lattices have so far been unsuccessful. The observed rearrangements cannot be unambiguously attributed to intrinsic allosteric effects. Our covariance analysis suggests that the Q256 sidechain plays a role in the structure of this important region of the protein that is disrupted upon mutation to alanine; at present, this remains conjecture.

**Variant S284A.** S284 sits at the third covariance hotspot we studied, a hydrophobic zipper formed by three residues (S284, A285, and L286) that bridges the dimer interface (Figs. 1G and 4E). This zipper has been highlighted as a key dimer interaction in previous work and is one of the few sequence positions that differ between M^pro from SARS-CoV-2 and SARS-CoV-1, where the sequence is STI instead of SAL[18,39]. This indicates that alanine is well tolerated at this position and may not affect the enzyme structure substantially, even if this location is an important covariance hotspot. Nonetheless, we elected to study the S284A mutant to maintain a consistent strategy with our other mutants.

The S284A mutant shows little change in activity when compared to wild type, and a comparable dimer affinity (Fig. 5 and Table 2). The crystal structure of this variant confirms that the mutation to alanine has little effect on the structure of M^pro, with an overall heavy-atom RMSD to a wild-type reference structure (PDB: 7AR6) of 0.75 Å. The hydrophobic zipper in the S284A variant exhibits a slightly larger protomer-to-protomer distance, with the A285/A285' distance increasing from 4.8 to 5.5 Å ($C_\alpha$, PDB 7AR6, S284A, respectively). We note these distances are comparable to the structural heterogeneity observed in our large set of wild-type structures (Supplementary Fig. 2).

We conclude that substitution of S284 for alanine has no significant effect on the protein structure, and in turn, no effect on function. Therefore, the degree to which a perturbation at this site would impact dimer activity remains undetermined and requires the study of additional mutants or other sources of perturbation.

**Of the three alanine mutants, one is a confirmed covariance hotspot.** Based on our covariance model, we predicted our mutants would exhibit both a reduction in monomer/dimer affinity and a reduction in activity of the residual dimers. Experiments on N214A confirmed this prediction; this position is an alanine-confirmed allosteric hotspot. Q256A shows a modest reduction in dimer and substrate binding affinities. Finally, mutating S284 to alanine failed to significantly perturb the protein structure, and the effect of perturbation at this position remains to be studied. These sites remain putative covariance hotspots.

**Structural diversity is a function of crystal lattice perturbation**
Our prediction of covariance hotspots at positions N214, Q256, and S284 relied on correlations inferred from structural variation in our diffraction dataset. Our hypothesis is that this variation reflects an important subset of the protein's solution-phase fluctuations, where the protein is buffeted by thermal motion of the solvent.

Interested in the origins of this variation, we discovered that the protein structures we determined were a function of the lattice parameters of the

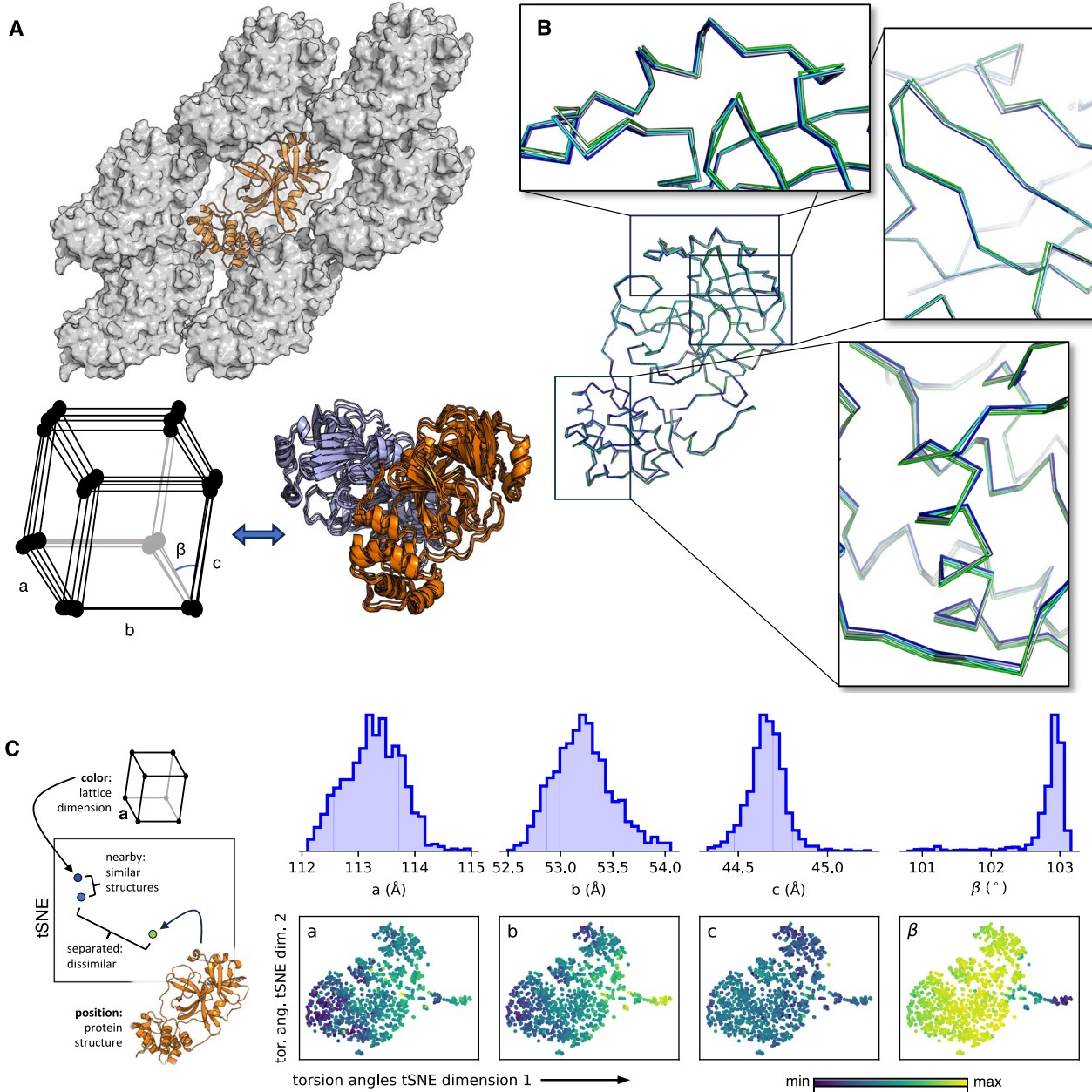

**Fig. 6 | Variation in M^pro crystal structures is a function of crystal lattice deformation. A** The atomic structure of the protein influences the crystalline lattice and vice versa. M^pro (orange ribbon) is shown along with the lattice (grey surface) for an exemplar structure, suggesting how changes in the protein conformation can produce changes in the crystal packing (cartoon below). **B** Continuous deformation of the M^pro structure is observed as a function of the unit cell volume. Five refined structures from our set of 1146 were sorted by unit cell volume (min: 256.3 nm³, blue/ max: 270.9 nm³, green) and drawn as a backbone ribbon. The highlighted regions show continuous deformation as a function of the cell volume. These regions are in the vicinity of crystal contacts, *cf.* (**A**). **C** Statistical view of the link between lattice parameters and determined atomic structure. Atomic structures were embedded into a two-dimensional space by performing t-SNE[86] on the structure's backbone torsion angles (φ, ψ), producing a visualization of the 1146 structures (see "Materials

and Methods"). As illustrated in the leftmost panel, along the bottom row, each point is a single atomic model of M^pro from a single crystal dataset; points that are nearby one another have similar structures (torsion angles), while distant points are dissimilar. Points are then colored based on the value of the indicated lattice parameter (blue: smallest, yellow: largest). Therefore, a smooth variation in color across the plotted points, as seen for lattice parameters *a*, *b*, and *β* indicates that the lattice parameters are correlated with the structure, though this connection is complex; a spatially random coloring, as seen for lattice parameter *c*, implies no strong relationship. While the lattice is clearly a determining factor of the final refined structure, other possible factors independent of the lattice cannot be excluded. The distribution of each lattice parameter over the entire set of 1146 structures is shown in the top row.

crystals they originated from (Fig. 6 and Supplementary Fig. 14). Specifically, the protein structure deforms in a smooth way as a function of lattice parameters (*a*, *b*, *c*, and *β* in our monoclinic space group), implying that deformations of the crystal lattice are propagated through deformations of the individual protein structures that form that lattice. In this model, as the lattice changes, the crystal contacts pull on one another—and thereby

transmit the strain on the lattice through the rest of the protein. The variation in structures observed is a direct product of this force. The fact that crystal contacts have high covariance with the active site (Fig. 3C) supports this idea. This does not imply that all protein structural variation can be accounted for by the lattice parameters, only that the lattice is strongly correlated with the final obtained structure.

Having established this link between lattice and structure, we investigated the primary causes of the lattice variability in our dataset. During our ligand screening effort, crystal samples were prepared by 14 different scientists, who fished crystals onto Kapton or nylon loops and flash-froze samples in liquid nitrogen. Because the fishing and freezing process exposes samples to dehydrating air for variable amounts of time, we reasoned that different crystallographers might work at different speeds, and the resulting spread of crystal dehydration would be a key factor introducing structural heterogeneity into our data.

To test this hypothesis, we opened a single vapor diffusion well to the dry lab atmosphere and fished multiple crystals as a function of time (Supplementary Fig. 15). Crystals were therefore able to dehydrate in situ in a 500 nL (set volume) sitting drop and fished at intervals spanning 5 to 205 s. Crystals fished at longer delays failed to diffract to high resolution. The crystals were then subjected to X-ray analysis, and the resulting series of structures confirms that air exposure results in a continuous contraction of the crystal lattice, accompanied by deformation of the $M^{pro}$ structures obtained. The total volume occupied by the protein in the lattice is constant, while the solvent content drops as a function of air exposure, as might be expected. We conclude that differential air exposure during fishing and freezing induces significant heterogeneity in both the crystal lattice and structure we obtain. We have not yet systematically investigated other perturbative forces, such as cryo-freezing (including freezing rate), crystal mounting, or the weakly interacting ligands present during crystal growth, but our experiments show hydration changes due to fishing speed alone are insufficient to explain the variation we observe (Fig. 6).

## Discussion

Our work extends crystallography beyond the determination of a single structure to the determination of structural distributions. By determining one such distribution, sampling the native state fluctuations of $M^{pro}$, we were able to gain insight into the enzyme's function. Our approach relies on studying an unprecedented number of crystals subject to perturbation and therefore relies on the intense research effort on $M^{pro}$ motivated by the COVID-19 pandemic. New technology, however, most notably serial crystallography and high throughput fragment screening—perhaps in combination with methods for inducing systematic perturbations—could be readily developed to produce similarly rich datasets on the order of a day with an effort manageable by a typical research group. This would establish *statistical crystallography*, defined as determining a structural distribution from a set of crystallographic data. This contrasts statistical crystallography from the usual crystallographic goal of obtaining a single representative set of atomic coordinates from a single dataset.

Such a method relies on perturbations that produce structural variability. While the perturbations we studied were serendipitous, the idea of actively perturbing crystals to gain information goes back to the birth of macromolecular crystallography, at first with the intent to solve the phase problem[43]. The idea of varying hydration to gain functional insight followed[44–46], and more recently, studies varying humidity have been joined by others using different crystal lattices, temperature changes and electric fields to study the structural distributions of myoglobin, CypA, lysozyme, and $M^{pro}$ itself[5,47–50]. Unlike our work, however, these studies employed datasets on the order of 10 crystals and made use of proactive perturbations. Kidera et al. and Saeed et al., in contrast, analyzed large numbers of $M^{pro}$ structures that were publicly available in a statistical fashion and found that these sets of structures allowed them to gain insight into the protein's structural fluctuations and function[26,51]. As all these research efforts analyzed sets of crystallographic data to understand distributions of structure, under our definition, they are statistical crystallography studies. Along with our contribution, these studies demonstrate that the crystal lattice is flexible enough to support continuous changes to the structure, and if subjected to enough random perturbations, can produce a conformational distribution. Evidenced by these studies, this distribution can capture harmonic backbone displacements, sidechain rotations, and near-native stable conformers compatible with the crystal lattice. Fluctuations that are either so large they disrupt the crystalline order or so small they cannot be seen at a given crystallographic resolution will be invisible. Further, while methods to predict function from such conformational distributions require additional development, our work endeavored to make and test falsifiable predictions of how modifications to sequence would impact function, specifically turnover kinetics.

Given this history and the results presented here, we propose the following perturbative crystallographic conjecture: that non-specific perturbations to protein crystals can reveal new, functionally relevant structural information about the crystallized protein. The free energy of biological ligand or substrate binding is generally small, on the order of a few $kT$. Thus, perturbations of this order of magnitude can spontaneously shift the protein from one conformational state into another, whether the protein is in a crystalline or solution state. In crystals, such changes will be observable if they do not irrevocably disrupt the order of the crystal lattice. There is, of course, no a priori guarantee that any single perturbation will yield a functionally insightful result, or that any specific crystal system will be able to support all fluctuations of interest. Our conjecture is simply that this will typically be the case; demonstrating or falsifying this requires the study of additional protein crystal systems. We do not expect the distribution of structures in crystals to be identical to those in solution—but just as static crystal structures are invaluable for understanding proteins' solution-phase function, we do expect that the dynamical information provided by statistical crystallography will enable new insights into protein function.

Our work on the $M^{pro}$ from SARS-CoV-2 forms a first case study testing the perturbative crystallographic conjecture and its implications for our ability to understand biological function, such as allostery. Our covariance model, based on the perturbative conjecture, identified an allosteric network connecting the dimer interface and active site via N214. Mutation at this hotspot position reduces $k_{cat}$, beyond what is expected from a trivial effect due to reduced dimer affinity. Further crystallography shows that this mutation disrupts catalytically active conformations of the essential oxyanion loop. Therefore, as predicted, N214 meets our definition of a covariance hotspot in $M^{pro}$. Further, since binding of a second $M^{pro}$ monomer at this site affects regulation, this covariance hotspot is an allosteric site.

These findings support previous work showing N214 plays a key role in regulating the enzymatic activity of $M^{pro}$ as a function of dimer formation. Notably, Song and colleagues showed via an alanine scan of SARS-CoV-1 $M^{pro}$ that mutating N214 to alanine deactivated the enzyme[39–41]. In contrast to our characterization of SARS-CoV-2 $M^{pro}$, they determined that for the SARS-CoV-1 enzyme, the dimerization constant increased only 2.5× upon mutation, and that the crystal structure of N214A remained largely unchanged as compared to wild type[40]. Later, by analyzing the set of available SARS-CoV-2 $M^{pro}$ crystal structures[26], Kidera and colleagues pinpointed the same N214/N-terminal hydrogen bond network our mutational studies identified; our crystal structure of the N214A variant confirms their hypothesis that this network is essential for the regulation of $M^{pro}$ by dimerization. Combined with our results, this work supports the idea that multiple crystal structures can, in favorable cases, reveal correlations with atomic detail.

In contrast, the Q256 and S284 positions were not confirm to regulate $M^{pro}$ *via* dimerization, and remain putative covariance hotspots. The Q256A variant did produce a modest reduction in enzyme activity, accounted for by a modestly reduced monomer/dimer $K_D$ and slightly compromised ligand binding affinity. This is non-trivial, as Q256 does not form any clear dimer-spanning interaction. At present, the precise structural consequences of making this mutation are unclear, as crystallization of this mutant results in a dramatic change in crystal packing. In contrast, our crystal structure of S284A shows that alanine in this position does not change the protein's structure significantly. Therefore, if S284 is structurally linked to the active site residues as we predict, our mutation will not perturb the active site structure or function, precluding an incisive test of this site as a covariance hotspot. More work on the structure and dynamics of $M^{pro}$ at these positions will be necessary to understand these subtleties.

Moreover, our present approach cannot comprehensively identify all residues known to form critical cross-dimer interactions. For example, R298 forms a salt bridge known to stabilize the M$^{pro}$ dimer[38,52] but does not appear prominently in our covariance analysis. A manual analysis of this residue shows that in our crystal's monoclinic lattice, the backbone of this residue is static, while the sidechain—which our analysis is not sensitive to—is only partially ordered (Supplementary Fig. 16). In other lattice packings, however, the sidechain becomes ordered, but the backbone varies from structure to structure. Therefore, to capture a more comprehensive set of covariance interactions, future efforts should study multiple crystal packings and aim to develop algorithms that can efficiently capture sidechain information.

For instance, we highlighted similarities between our crystallographic data and molecular dynamics simulation as a point of strength. MD simulations can model proteins in solution, without the potential caveat of restrictions due to lattice confinement. On the other hand, MD relies on classical force fields that may introduce error and may be limited by sampling in the cases of large systems or many variants of interest (mutants, ligands, etc.). We analyzed three distinct MD simulations of M$^{pro}$ that had been previously published, and found that while broadly similar, the predicted atomic covariances differed between them in meaningful ways (Supplementary Figs. 5 and 7). In light of these important differences, experimental techniques such as statistical crystallography—as well as established NMR and emerging cryoEM methods to access distributions of structure—are a necessary complement to simulations.

The routine collection of thousands of crystal structures, under perturbation, is within reach of modern hardware[4,5,49,53]. At synchrotron sources, collection rates of about a dataset per minute are now typical, which means thousands of datasets can be collected per day. New methods for automated crystal fishing and freezing will be necessary to take advantage of this capability, but such developments in crystal handling using photoablation and acoustic fields[54] are underway. Alternatively, serial crystallography at room temperature provides an established and facile route to the study of up to millions of crystals in a single experiment[55]. In the presence of perturbations due to external fields[5], humidity[56], temperature[53], or acoustic waves[57], room-temperature serial experiments could provide a second route to routine statistical crystallography. Whether serial or rotational data collection is employed, using random perturbations may enable the inclusion of large sets of data, while controlled perturbations are almost certain to provide more efficient exploration of conformational space on a per-crystal basis.

Software advances will also be essential to enable statistical crystallography. First, new methods for fully automated crystallographic refinement are urgently needed, and machine learning methods are rapidly advancing to meet this challenge[58]. Second, our statistical work focused only on the covariance of C$_\alpha$ atoms. Important sidechain rotations, sidechain-backbone coupling, and non-Gaussian displacements of the backbone remain to be analyzed (see Supplementary Figs. 3, 4, and 16), and for now preclude the discovery of allosteric interactions such as those mediated by R298 that involve sidechain motion without significant accompanying backbone displacement. We expect new techniques for analyzing sets of protein structures—originating from crystallography or other sources—will enable us to uncover more complex patterns of information transfer through the structure of M$^{pro}$ and other systems. Third and finally, methods that can quantitatively use information about protein distributions to make quantitative predictions about function, such as enzyme catalytic rates, will constitute a major breakthrough. We expect that work studying crystal structures under perturbation will facilitate the formulation, training, validation, and use of models that map distribution to function.

We inferred a model of atomic correlations from the study of thousands of crystals, which had undergone random perturbations during sample preparation. We then used our model to make specific predictions about protein function and performed experiments to test those predictions, uncovering a key regulatory network in M$^{pro}$ mediated by N214. Our study analyzed a set of crystallographic datasets to understand conformational flexibility. This statistical crystallography provides a general method to understand the structural fluctuations that enable protein function, a task that has been highlighted as a grand challenge in biophysics[59].

## Materials and methods
### Expression and purification of SARS-CoV-2 main protease mutants
The wild-type sequence and three-point mutants (N214A, Q256A, and S284A) of the SARS-CoV-2 main protease (M$^{pro}$) were cloned and inserted into the pGEX-4T-1 vector (GeneWiz). Following previous designs[18], the insert contains, in order: an N-terminal GST tag, M$^{pro}$ cleavage site, the M$^{pro}$ sequence, an HRV C3 cleavage site, and a C-terminal His$_8$-tag. To express recombinant protein, plasmids were transformed via heat shock at 42 °C into *E. coli* BL21 DE3. Strains were then pre-cultured in LB supplemented with 100 μg/mL ampicillin overnight at 37 °C, then inoculated into fresh media at a 1:100 ratio and cultured at 37 °C. At an OD$_{600}$ of 0.6, overexpression was induced via 0.5 mM IPTG and the temperature was decreased to 18 °C. The cells were harvested after 16 h and frozen at −20 °C.

For purification, the cell pellet was resuspended in Tris-HCl/NaCl (20/150 mM) buffer at pH 7.5 and lysed by sonication. Lysate was clarified by ultracentrifugation at 120,000 × *g* for 30 min and the supernatant was filtered through 0.8 and 0.22 μm membranes. Clarified lysate was then purified by Ni-affinity with a HisTrap HP and rinsed with 40 mM imidazole prior to elution of the protein of interest using 250 mM imidazole. Following this, buffer exchange to Tris-HCl/NaCl/TCEP (20/150/1 mM) was performed with a PD-10 desalting column, and the His$_8$-tag was cleaved by the addition of HRV C3 protease (10 units per mg protein, overnight at 4 °C). A second passage over a HisTrap HP (reverse IMAC) removed contaminants with metal affinity. Finally, the protein was concentrated and polished via SEC (Superdex 75, Hiload 16/600) with Tris-HCl/NaCl/TCEP/EDTA (20/150/1/1 mM) running buffer. Identity and purity of the protein-containing fraction were confirmed by SDS-PAGE; peak fractions were concentrated, flash frozen in liquid nitrogen, and stored at −80 °C.

### Analytical size exclusion chromatography
Analytical size exclusion chromatography (SEC) was performed with an ÄKTA pure system (Cytiva) employing a Superdex S75 Increase 10/300 column at room temperature, approximately 22 °C. The column was equilibrated with TrisHCl/NaCl/TCEP/EDTA (20/150/1/1 mM) buffer. Fifty microliters of the wild-type or mutant M$^{pro}$ were run at 25, 12.5, 6.25, 3.1, 1.6, and 0.8 μM. For N214A, additional measurements at 100 and 50 μM were collected to sufficiently span the increased monomer/dimer $K_D$.

The wild type, Q256A and S284A variants ran as two distinct peaks, which were assigned to dimer and monomer species. This observation implies the equilibrium kinetics, in which monomer and dimer species interconvert, are slow on the timescale of the experiment (~15 min). The fraction of monomer species $R = [M]/([M] + 2[D])$ was quantified by fitting a Gaussian to each peak and comparing the ratio of areas under the curves. A final $K_D$ was then determined by fitting the binding curves (Fig. 5B) to a monomer-dimer equilibrium, $2M \rightleftharpoons D$. Under the assumption that the equilibrium kinetics are slow

$$R = \frac{\sqrt{8E_0/K_D + 1} - 1}{4E_0/K_D} \qquad (1)$$

where again $E_0$ is the total concentration of M$^{pro}$ polymer chains. Using this relationship, $K_D$ was fit by non-linear least squares.

In contrast, the N214A variant ran as a single, skewed peak, implying that the equilibration kinetics are rapid on the timescale of the experiment. In this regime, the relative concentration of monomer can be roughly estimated by the elution volume of the single peak. For the N214A variant, therefore, this method was used instead of the area under the Gaussian peaks to determine the ratio of monomer and dimer species, which were then fit to determine a $K_D$. For all SEC runs, the nominal concentrations were corrected by integrating the area under the absorbance trace at 280 nm.

## Isothermal titration calorimetry (ITC)

All experiments were performed with a MicroCal PEAQ-ITC (Malvern Panalytical). Protein was dialyzed overnight into HEPES/NaCl/TCEP/EDTA/DMSO (30/150/1/1 mM/3%), then diluted to 20 µM. Final protein concentrations determined from the ITC data can be found in Supplementary Table 2. The ligand calpeptin was solubilized in 100 % DMSO, then diluted to a final concentration of 400 µM with dialysis buffer. The ligand was titrated from the syringe into the cell filled with protein solution (approximately 20 µM, for precise values see Supplementary Table 3). All measurements were performed at 28 °C with a reference power of 10 µcal/s, low feedback mode enabled, stirring at 750 rpm, an initial delay of 60 s, an injection spacing of 150 s, and an injection duration of 4 s. In total, 19 injections were performed; the first injection point was discarded from the analysis. First injection volume was 0.4 µL, all following were 2 µL. The data were fit and analyzed with the MicroCal PEAQ-ITC Analysis Software v1.41 (Supplementary Table 2 and Supplementary Fig. 6).

## Nano differential scanning fluorimetry (nDSF)

Protein samples were prepared in the same way as for ITC measurements, with final protein concentrations of 10 µM for wild type and 14 µM for the N214A variant. Calpeptin was solubilized in 100% DMSO, then serially diluted to concentrations from 1 mM to 244 nM using dialysis buffer. Measurements were performed with a Prometheus NT.48 fluorimeter (NanoTemper Technologies) controlled by PR.ThermControl (NanoTemper technologies, version 2.1.2) for wild type and with a Prometheus PANTA (NanoTemper Technologies) controlled by PR.Panta Control (NanoTemper technologies, version 1.8) for $M^{pro}$ N214A. Temperature was ramped from 25 °C to 95 °C with a temperature slope of 1 °C per minute with excitation power of 98%. All measurements were performed in duplicate. Visualization of the data and isothermal analysis (Supplementary Fig. 11) were performed following the procedure developed by Niebling et al.[60] and Burastero et al.[61], specifically for the isothermal analysis of the melting curves, the 350 nm/330 nm fluorescence ratio in a temperature range between 40 °C and 65 °C was fit to a two-state model assuming zero $\Delta C_p$ for the unfolding. The calculated fraction unfolded vs. ligand concentration was then fit with a two-state binding model into obtain $K_D$ values at 52 °C, 53 °C and 54 °C.

## Native mass spectrometry

Using Micro Bio-Spin P-6 columns (Bio-Rad, 6000 MWCO) and performing two cycles, purified wild-type and mutant $M^{pro}$ were buffer exchanged into 300 mM ammonium acetate, pH 8.0 and 1 mM DTT, which preserves non-covalent interactions and is known to be compatible with $M^{pro}$[62]. All variants were measured at 2 µM. In addition, wild type was measured at 10 µM and N214A at 10 µM and 20 µM.

Samples were measured in triplicate with a Q-Exactive UHMR Orbitrap (Thermo Scientific) and introduced via gold-coated nanoESI capillaries. Capillaries were pulled in-house from borosilicate capillaries (1.2 mm and 0.68 mm outer and inner diameter, respectively, *World Precision Instruments)* using a two-step program with a micropipette puller (Sutter Instruments, P-1000). Gold-coating was done by using a sputter coater (CCU-010, Safematic, $5.0 \times 10^{-2}$ mbar, 30.0 mA, 120 s, three runs to vacuum limit $3.0 \times 10^{-2}$ mbar argon). Native MS was performed in positive ion mode by applying capillary voltages of 1.5–1.7 kV, 50 °C capillary temperature, 10 V in-source CID and 15 V in the HCD cell. Trapping gas pressure optimization was set to 5. Ultra-high vacuum was ~$2.6 \times 10^{-10}$ mbar. The detector settings and the ion transfer m/z optimization were set to "low m/z".

Each observed peak was assigned to either monomer or dimer species. Collision-induced dissociation showed that the only overlapping peak ($M^{9+}$/$D^{18+}$) is predominantly monomeric. Areas under the curve were estimated, summed and normalized to determine the relative intensities of monomer and dimer species, $I_M$ and $I_D$, respectively. These intensities are proportional to the concentrations of their respective species. Then, the absolute concentration of monomer is given by

$$[M] = \left(\frac{E_0}{I_M + 2I_D}\right) I_M$$

and dimer by

$$[D] = \left(\frac{E_0}{I_M + 2I_D}\right) I_D$$

where as before $E_0$ is the total concentration of enzyme molecules in either monomer or dimeric form, which was determined by UV absorption at 280 nm offline.

$K_D$s were estimated by fitting equation [1] with least squares and, alternatively, averaging measurements at each concentration to rule out concentration-dependent systematic errors (Supplementary Table 4). For the latter approach, $K_D = [M]^2/[D]$ was computed on a per-measurement basis, then measurements of the same concentration were combined to compute an average $K_D$ along with corresponding standard errors of the mean. We elected to report the averaged measurements at 2 µM, as these values are expected to suffer from the least systematic error due to ion clustering or suppression. The mass spectrometry proteomics data are available via ProteomeXchange with PRidentifier PXD076496.

## Enzyme activity assay

Fluorescent assays were performed on a CLARIOstar Plus plate reader (BMG Labtech) with 96 well UV transparent half flat bottom microplates (Corning). The substrate Dabcyl-K-TSAVLQSGFRKM-E-EDANS-NH$_2$ was dissolved in 100% Cyrene and diluted to the desired values for the experiment. Enzyme, in Tris-HCl/NaCl/TCEP/EDTA/Cyrene (20/150/1/1 mM/10%), was mixed with the substrate in a 1:1 ratio inside the wells to initiate the reaction. Final substrate concentrations were 320, 160, 80, 40, 20, 10, and 5 µM, while enzyme concentrations were 8, 4, 2, 1, 0.5, 0.25, and 0.1 µM, with additional measurements of 256, 128, 64, 32, and 16 µM for the N214A variant. Fluorescence intensity data, using filters for excitation of 360 ± 15 nm and emission of 460 ± 20 nm, were collected from 240 cycles of data collection with cycle interval times of 15 s. Detector gain was fixed manually to 1400. All measurements were performed at 25 °C.

We corrected for the inner filter effect, in which a high concentration of absorbing substrate decreases the observed fluorescence from cleaved product[63]. To do so, we measured the fluorescence of EDANS at a range of concentrations in the presence of a series of concentrations of labeled substrate peptide. The resulting fluorescence data were fit to an empirical model derived under the following assumptions: that (i) the EDANS-coupled peptide product and free EDANS have the same extension coefficient (15,100 M$^{-1}$ cm$^{-1}$ at 472 nm), (ii) the substrate fluorescence is linear in concentration and independent of the concentration of product, (iii) the product fluorescence is linear in concentration prior to attenuation by the inner filter effect, and (iv) the inner filter effect i.e., the reduction in product fluorescence is a power law as a function of background substrate. The power law relationship was determined empirically after inspecting data from our plate reader, following attempts to fit the data to the phenomenological absorbance-based models of Lakowicz[64] and Huyke et al.[65], neither of which produced satisfactory results. Together, these assumptions imply

$$f = \left(a[S]^b - c\right)[P] + d[S] + e$$

where $f$ is the observed fluorescence, $[P]$ is the concentration of EDANS or product, $[S]$ is the substrate, and $a, b, c, d,$ and $e$ are fit scalar parameters. The parameters were determined from data taken at substrate concentrations of 640, 320, 160, 80, 40, 20, 10, and 5 µM. For each of these concentrations, a background-subtracted fluorescence reading was measured for EDANS concentrations of 0.0, 2.5, 5.0, 7.5, and 10.0 µM. Parameters were fit using non-linear least squares. Subsequently, this model was used to correct for the concentration of the product.

Following inner filter correction, the initial velocities, $V_0$, were estimated for each enzyme/substrate concentration by fitting a line to the inner-filter corrected fluorescence data by linear least-squares. As the reaction velocities varied considerably for different enzyme/substrate concentrations, the linear region to fit was determined algorithmically by scanning the fit region from the first 90 seconds of data collection up to 32 min of data collection. The highest velocity fit was reported. This simple procedure corresponds well with fits that would have been manually curated, as it automatically selects a high-velocity linear section of the data and increases the number of fit datapoints to minimize the effects of outliers.

We considered three approaches to modeling the resulting data (Supplementary Fig. 12A). In the simplest, we fit every enzyme concentration with an independent Michaelis–Menten model (Fig. 5 and Supplementary Fig. 12B),

$$V_0 = \frac{k_{sp}E_0[S]}{1 + k_{sp}[S]/k_{cat}}$$

where $k_{sp} = k_{cat}/K_M$. Using $k_{cat}$ and $k_{sp}$ as fit variables, rather than $k_{cat}$ and $K_M$, produces a more robust fit and facilitates error estimation of $k_{cat}/K_M$, which has a more direct physical interpretation than $K_M$ alone[66]. This approach employs a large number of fit parameters, and the $k_{cat}$ and $k_{sp}$ values represent averages over the monomer and dimer species present at any single enzyme concentration, however we favored it due to the straightforward physical interpretation.

Alternatively, we considered a model where two $M^{pro}$ monomers and a single substrate molecule come together to form an active enzyme-substrate complex, which is assumed to be at steady-state concentration early in the reaction course (Supplementary Fig. 12C). The rate of enzyme-substrate complex formation is therefore second order in free enzyme concentration. This model is the simplest we considered, parameterized by just two fit values, an analog of $K_M$ we simply call $K_{eq}$, and $k_{cat}$, with the final initial velocity given by

$$V_0 = \frac{k_{cat}}{8}\left[4E_0 + K_{eq}/[S] - \sqrt{\left(4E_0 + K_{eq}/[S]\right)^2 - 16E_0^2}\right]$$

This model provides the right qualitative behavior to explain the assay data for each variant using just two fit parameters, but does not provide a satisfactory fit to the observed data (Supplementary Fig. 12C).

Finally, we fit a model in which $M^{pro}$ exhibits a slow monomer/dimer equilibration, the monomer is negligibly active, and the dimer exhibits Michaelis–Menten kinetics (Supplementary Fig. 12D). In this model, the concentration of dimer species is

$$[D] = E_0 - \frac{\sqrt{8E_0/K_D + 1} - 1}{4/K_D}$$

where $E_0$ is the total enzyme concentration in either dimer or monomer form, $E_0 = 2[D] + [M]$. The dimer species then turns over with Michaelis–Menten kinetics. This model provides a satisfactory fit of the biochemical data for all variants besides N214A, which hindered interpretability.

All models were fit by nonlinear least squares, with Gaussian errors propagated from the linear steady-state fits through to the final Michaelis–Menten parameters. These confidence intervals account for statistical error only and do not report systematic errors introduced by the experimental or analysis procedure.

## Crystallization and diffraction data collection

Ligand-free, wild-type crystals were grown as part of a ligand screening campaign, previously reported[31]. Prior to crystallization, 125 nL droplets of 10 mM compound solutions in DMSO were applied to the wells of SwissCI 96-well plates (2-well or 3-well low profile) and subsequently dried in vacuum. Co-crystallization with screened compounds was achieved by mixing 0.23 μL of protein solution (6.25 mg/mL) in 20 mM HEPES buffer (pH 7.8) containing 1 mM DTT, 1 mM TCEP, 1 mM EDTA, and 150 mM NaCl with 0.22 μL of reservoir solution consisting of 100 mM MIB (pH 7.5) containing 25% (w/w) PEG 1500 and 5% (v/v) DMSO, and 0.05 μL of a micro-seed crystal suspension using an Oryx4 pipetting robot (Douglas Instruments). This growth solution was equilibrated by sitting drop vapor diffusion against the reservoir solution, resulting in 100 μm to 200 μm plate-like crystals after 2 to 3 days.

Crystals for the $M^{pro}$ variants N214A, Q256A, and S284A were grown via sitting drop vapor diffusion at room temperature, in which 1 μL of protein solution (6 mg/mL) was mixed with 1 μL of reservoir solution and equilibrated against the reservoir solution. This resulted in plate-like crystals with sizes ranging from 100 to 200 μm² after 3 to 5 days. The reservoir solution for $M^{pro}$ N214A contained 0.12 M ethylene glycols (diethylene glycol; triethylene glycol; tetraethylene glycol; pentaethylene glycol); 0.1 M Morpheus buffer system 3 (1.0 M Tris, 1.0 M bicine, pH 8.5) and 50% (v/v) Morpheus Precipitant Mix 1 (40% (v/v) PEG 500 MME, 20% (w/v) PEG 20,000). For $M^{pro}$ Q256A, it contained 20% (w/v) PEG 1500, 0.1 M MMT at pH 7.5, and 5% ethylene glycol and for $M^{pro}$ S284A, 27.5% PEG 1500, 5% DMSO, and 0.1 M MIB at pH 7.5. No additional cryoprotectants were used.

Crystals of the Q256A and S284A mutants grew in conditions similar to those used for the wild type; while S284A exhibits the same monoclinic lattice as the wild type, the crystal packing of Q265A forms a distinct orthorhombic lattice. In contrast, the N214A variant did not crystallize in similar conditions. Following condition screening from scratch, a new crystal system was developed that enabled the growth of crystals that diffracted to high resolution in a different orthorhombic lattice from Q256A, but one that has been observed for $M^{pro}$ previously[67]. The condition contains a mixture of ethylene glycols, which were observed bound to the protein in the resulting structure.

For both mutant and wild type, crystals were mounted and flash-frozen in liquid nitrogen. Diffraction experiments were performed at PETRA-III beamline P11 (DESY, Hamburg), delivering a 100 μm beam of 12 keV X-rays focused by a paired KB mirror system[68]. Crystals were mounted robotically on a single-axis goniometer and held at 100 K using a cryojet (Oxford). During data collection, samples were rotated 200° with frames read out from a DECTRIS Eiger detector every 0.2° (wild type and Q256A mutant) or 0.1° (N214A and S284A).

For crystals fished as a function of time, we reproduced the crystallization conditions used in our ligand screen[31] but omitted any ligands. Specifically, crystallization was achieved by mixing 0.23 μL of protein solution (6.25 mg/mL) in 20 mM HEPES buffer (pH 7.8) containing 1 mM DTT/TCEP (respectively), 1 mM EDTA, and 150 mM NaCl with 0.22 μL of reservoir solution consisting of 100 mM MIB, pH 7.5, containing 25% (w/w) PEG 1500 and 5% (v/v) DMSO, and 0.05 μL of a micro-seed crystal suspension using an Oryx4 pipetting robot (Douglas Instruments). This growth solution was equilibrated by sitting drop vapor diffusion against 40 μL reservoir solution. A timer was set, and crystals were manually harvested and cryo-cooled in liquid nitrogen at specified intervals for subsequent X-ray diffraction data collection.

## Automated refinement of ligand-free datasets from the DESY screening campaign

As previously reported[31], an automatic data processing and structure refinement pipeline *xia2pipe* was employed to automatically refine crystals from our ligand screening campaign, including the ligand-free crystals reported here. Raw diffraction images from the PETRA III beamlines were processed using three crystallographic integration software packages: *XDS*[69], *autoPROC*[70] followed by *staraniso*[71], and *DIALS* via *xia2*[72,73]. The results of each dataset were then automatically refined using *phenix*[74]. Refinement began by choosing one of two manually refined starting models based on a best match of unit cell parameters, followed by (i) rigid body and ADP refinement, (ii) simulated annealing, ADP, and reciprocal space refinement, (iii) real-space refinement, and (iv) a final round of reciprocal space

refinement as well as TLS refinement, with each residue pre-set as a TLS group. Ligand-bound datasets were identified[31] and discarded in this work. Further, crystals grown in the presence of any identified hit or positive control ligand were discarded. Finally, datasets were selected based on resolution (2.2 Å or better) and refinement quality ($R_{free}$ at least 0.25 or lower), producing 1146 final structures.

### Determination of mutant crystal structures
The N214A, Q256A and S284A datasets were processed with *DIALS*[75,76]. Strong anisotropy was observed in the N214A data; accordingly, this dataset was merged and truncated with *staraniso*[71]. A summary of the data statistics is given in Table 1. Resolution limits were determined for N214A by *staraniso* using $I/\sigma_I > 1.2$ or by *DIALS* using $CC > 0.3$. The Q256A dataset was further truncated due to diffraction beyond the edge of the detector. Structures were solved by molecular replacement using *PHASER*[77] with 7AR6[31] as a search model for the S284A variant and 7BB2[67] for the N214A and Q256A mutants. Final models were determined with iterative rounds of manual building using *coot*[78] and refinement using *phenix*[74]. The resulting models were validated using *MolProbity*[79].

### Estimation of covariance matrices and covariance with the active site
$C_\alpha$ covariances were estimated from the set of ligand-free structures as follows. The 1146 M^pro structures were centered by subtracting their centers of mass from each atom position. Subsequently, the isotropic $C_\alpha$ covariance between atoms $i$ and $j$ was computed as

$$C_{ij} = 8\pi^2\left[\left\langle\delta x_i \delta x_j\right\rangle + \left\langle\delta y_i \delta y_j\right\rangle + \left\langle\delta z_i \delta z_j\right\rangle\right]$$

where $\delta x_i$ is the mean-subtracted position of the x-component of the Cartesian position of atom $i$, and the other quantities are defined analogously. Angle brackets indicate an average over all structures. Finally, the factor of $8\pi^2$ ensures the scale is the same as is typically used for crystallographic B-factors. To provide an idea of the statistical variability of this estimate, a 1000-sample bootstrap was employed[80]. Specifically, for each sample, 1146 structures were drawn randomly with replacement from the pool of 1146 structures, and this sample was used to compute residue-residue covariances. The variability in this covariance estimate was then quantified by the standard deviation across these 1000 bootstrap samples. To estimate the covariance with the active site, we averaged over residues 41, 49, 143–145, 163–167, 187–192, following the active site definition of Chen et al.[81].

### Estimation of alternative measures of correlation
For canonical correlation analysis (Supplementary Fig. 3), the implementation in *scikit-learn* was employed[82]. Mutual information (Supplementary Fig. 4) was estimated using the *mdentropy* package[83]. For the $C_\alpha$ mutual information, we used the *knn* algorithm with $k= 8$, while for the dihedrals, the *knn* algorithm with $k = 16$ and shuffling enabled was employed. The conditional displacement (Supplementary Fig. 4) reports the distance one atom moves conditioned on another moving a specified distance, assuming the displacements are distributed as a multivariate Gaussian. Specifically, conditioned on one atom moving by distance $r_2$, the conditional displacement is given by the ratio

$$\frac{r_1}{r_2} = \sqrt{\frac{1}{3}Tr\left\{\Lambda^{-1/2}\right\}}$$

where $r_1$ is the distance another atom is expected to move and $\Lambda = \Sigma_{22}^{-1}\Sigma_{12}^T\Sigma_{12}\Sigma_{22}^{-1}$, with $\Sigma_{12}$ is the 3 x 3 matrix of covariances between the x, y, and z components of the positions of atoms 1 and 2, and $\Sigma_{22}$ is the covariance matrix for the corresponding values of atom 2 only. A detailed derivation, along with code to compute this quantity, can be found at [https://doi.org/10.5281/zenodo.19157701].

### Ensemble refinement
Ensemble refinement was performed using *phenix*[74,84]. First, the percent of atoms used in TLS analysis (pTLS) parameter was optimized using a single dataset; we tested values from 1.0 to 0.2 in increments of 0.1 and selected 0.6 based on the result that produced the minimum $R_{free}$. Then, we selected four diffraction datasets that evenly span the unit cell volumes present in our entire collection of crystals and conducted ensemble refinement on each with this pTLS. All other parameters were set to their defaults and the starting model used was the automatically refined result described above. Ensemble refinements are available *via* Zenodo [https://doi.org/10.5281/zenodo.13268794].

### t-SNE
t-SNE plots were used to assess if a link existed between the lattice parameters of a given crystal and the refined structure originating from that crystal (Fig. 6). To generate these plots, $(\varphi, \psi)$ torsion angles were featurized using sine and cosine of each angle. t-SNE was performed using PCA initialization and a perplexity parameter of 10. Points were colored by a given lattice parameter. As controls, we also performed the same procedure using PCA instead of t-SNE as a dimensionality reduction method (Supplementary Fig. 14A), which supports the conclusions of the t-SNE plots. Further, as a control, we randomly scrambled the lattice parameters, reassigning them to different structures in the dataset. We then re-performed the t-SNE analysis, which resulted in structureless plots (Supplementary Fig. 14B).

### Statistics and reproducibility
Covariance matrices were estimated from 1146 independently refined crystal structures, with uncertainty quantified by 1000-sample bootstrap with replacement. Native MS was performed in triplicate; nDSF in duplicate; SEC, ITC, and the fluorescent activity assay were each performed once per variant. Kinetic parameters were determined by nonlinear least squares with Gaussian errors propagated from steady-state fits; $K_D$ values from native MS are reported as means ± standard error across triplicate measurements.

### Reporting summary
Further information on research design is available in the Nature Portfolio Reporting Summary linked to this article.

## Data availability
Diffraction data, models, and maps for M^pro variants N214A, Q256A and S284A can be accessed at the PDB under accession IDs 9GI6, 9GHN, and 9GHO, respectively. Diffraction data, models, and maps for the 1146 ligand-free wild-type M^pro structures are available via Zenodo [https://doi.org/10.5281/zenodo.13268794]. The mass spectrometry proteomics data are available via ProteomeXchange with identifier PXD076496. Raw SEC data (Supplementary Dataset 1), SEC concentration fits (Supplementary Dataset 2), raw ITC and NDF data (Supplementary Dataset 3), fit kinetics velocities (Supplementary Dataset 4), and Michaelis–Menten parameters (Supplementary Dataset 5) are all available as supplementary data.

## Code availability
All code used to analyze data and generate figures, along with raw kinetic assay and SEC data, are available on github and archived to Zenodo [https://doi.org/10.5281/zenodo.19157701].

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

## Acknowledgements

We acknowledge the contributions of all members of the DESY COVID screening collaboration to the generation of the data analyzed here. Further, the authors would like to thank Helen Ginn for discussions and criticism, and Roberto Battistutta for an engaging exchange regarding R298 in particular. We thank Christian Günther for support with the nDSF measurements and David Ruiz Carrillo for support with the crystal screens, and the DESY P11 staff for their assistance with the crystallography experiments. Funding: T.J.L. was supported by a Helmholtz Young Investigator Award. We acknowledge financial support obtained from the Cluster of Excellence "Advanced Imaging of Matter" of the Deutsche Forschungsgemeinschaft (DFG)—EXC 2056—project ID 390715994, BMBF via projects 05K19GU4 and 05K20GUB, the Helmholtz society through the projects FISCOV and SFragX, and the Helmholtz Association Impulse and Networking funds InternLabs-0011 "HIR3X". This research was supported through computational resources (Maxwell cluster) and experimental facilities (PETRA III beamline P11) operated by Deutsches Elektronen-Synchrotron DESY, Hamburg, Germany, a member of the Helmholtz Association HGF. We acknowledge access to the Sample Preparation and Characterization (SPC) Facility of EMBL, Hamburg. C.U. and K.S.K. acknowledge funding through EU Horizon 2020 ERC StG-2017 759661 and EIC Pathfinder Open ARIADNE 964553. K.S.K. further acknowledges funding from RTG2771 Humans & Microbes, and C.U. further acknowledges support through BMBF VirMScan 13GW0622 and the City of Hamburg equipment grant.

## Author contributions

A.C., E.S. and T.J.L. conceived of the work, carried out experiments, analyzed data, and wrote the paper. P.R., A.R.M., S.G., S.N., K.S.K. and C.U. performed experiments, data analysis and provided feedback during writing. A.M., H.N.C., C.U. and T.J.L. secured funding and resources to support the work. J.S. developed the conception of the work and assisted in writing the paper. T.J.L. oversaw and administered the project.

## Funding

## Competing interests

The authors declare no competing interests.
