## [Transparent Peer Review File · Communications Biology]

Statistical crystallography reveals an allosteric network in SARS-CoV-2 M^{Pro}

Corresponding Author: Dr Thomas Lane

Version 0:

Reviewer comments:

Reviewer #1

(Remarks to the Author)

The manuscript "Statistical crystallography reveals an allosteric network in SARS-CoV-2 M^{Pro}" by Creon et al. presents an exceptionally large crystallographic dataset and introduces an innovative framework for extracting covariance information across perturbed crystal structures. The study is technically impressive and conceptually novel. However, several interpretative and methodological points require clarification to ensure that the conclusions are strong and not overstated.

1. The manuscript introduces "statistical crystallography" as a general framework but does not clearly define its limits. The authors should explicitly state which types of motions are accessible (e.g., low-energy elastic deformations compatible with lattice integrity) and which are not, to avoid overgeneralization.
2. Although lattice effects are discussed, but more quantitative comparison between crystallographic covariance and MD simulations. Explicit metrics distinguishing shared versus crystal-specific covariance would reinforce the argument that the identified networks reflect intrinsic protein behavior.
3. N214, Q256, and S284 are presented as equivalent covariance hotspots despite markedly different experimental outcomes. The authors should distinguish experimentally validated sites (e.g., N214) from putative or structurally tolerant sites and clarify that covariance alone does not necessarily imply functional sensitivity to alanine substitution.
4. In Supplemental Table 1, the S284A dataset shows an outer-shell $\langle I/\sigma(I) \rangle$ of 0.30 and CC1/2 of 0.179, values well below commonly accepted resolution cutoff thresholds. The manuscript does not justify inclusion of this shell. The authors should clarify the cutoff criteria used and indicate whether truncation at a more conservative threshold affects refinement statistics or conclusions regarding S284A.
5. The structural interpretation of Q256A remains ambiguous due to altered crystal packing. The authors should more clearly acknowledge that the observed rearrangements cannot be unambiguously attributed to intrinsic allosteric effects.
6. The analysis is restricted to C α covariance, although key allosteric and catalytic interactions in M^{Pro} are sidechain-mediated. The authors should explicitly justify this limitation and briefly discuss how sidechain contributions may be underrepresented.
7. Global R_{work}/R_{free} values do not guarantee local interpretability. Providing local map quality metrics (e.g., real-space correlation coefficients or representative density for key regions) would strengthen confidence that subtle structural differences are supported by experimental data.
8. The N214A mutant provides strong validation, but the mechanistic link between N214 disruption, N-terminal disorder, oxyanion hole destabilization, and reduced catalysis could be articulated more explicitly.

Reviewer #2

(Remarks to the Author)

The manuscript by Creon et al. builds on the authors' previous work where they performed crystallographic ligand screening, producing, among many ligand-bound SARS-CoV-2 M^{Pro} structures, a very large set of apo-structures. This set has now been utilized to down-select 1146 structures for structural analysis using the atomic displacement covariance as a measurable change in the enzyme conformations. This method, introduced as statistical crystallography, identified three covariance hotspots in the M^{Pro} sequence – residues that may be allosterically linked to the active site and influence the enzyme activity. Three point mutations have been introduced and the activity, dimerization and structures of N214A, Q256A

and S284A mutants have been studied. Interestingly, only N214A mutation demonstrated a significant effect on the enzyme dimerization (Kdimer), its activity and structure. This is an interesting result, but the presentation, interpretation and conclusions need to be improved and checked for factual accuracy.

The major issue is with the subsection on Variant N214A. The leading paragraph states that N214 side chain bridges the dimer interface by interacting with the N-terminus of the dimeric protomer. This statement is confusing because N214 H bonds with the main chain of G2 of the same protomer. Then, how does it bridge the dimer interface? Figure 5 does not explain this at all.

The text then goes on to talk about how N214A mutation disrupts the oxyanion hole, which made of residues 143, 144 and 145, not 139 through 145 as the authors claim, because only 3 amides can create an oxyanion hole. Here, again, if one double checks the structure, it becomes clear that N214 is 14A away from G143, so that it is hard to visualize how such a distant residue can disrupt the oxyanion hole. Thus, the H bonding network needs to be depicted for both the wt and N214A. Has the oxyanion hole actually unwound like in the monomeric structures (e.g. MPro1-199)? It is claimed that the oxyanion hole collapses – what does that mean?

The quality of the depicted electron density in Figure 5A and 5C is poor. Make the density more visible. In Figure 5C, does the N-terminus even have interpretable density?

The first paragraph of subsection Variant Q256A states that the C-terminus plays a significant role in forming the dimer complex. However, in many MPro structures the C-terminal residues beyond 302 are disordered, with sometime no detectable electron density. That means the C-terminus is disordered, supported by numerous MD simulations. How can it play a significant role in dimer formation?

The subsection “Three variants yield a single allosteric network” is rather dubious. How was the prediction made that mutants would exhibit reduction in activity – just covariance plots do not predict that. What is a single allosteric network? Either, rewrite this section to explain what this is about, or completely remove it.

The last paragraph before Discussion describes a very confusing, strange and poorly articulated exercise, including Figures 6 and S14. How were the crystals exposed to air – sitting in their drops for a while, right after mounting on a loop, after cryoprotecting? What were the crystal drop volumes for these experiments? Methods section doesn't appear to have any details. It is also hard to believe that drops of several microL in volume would lose significant amounts of water in just seconds to be observed in the unit cell contraction.

Reviewer #3

(Remarks to the Author)

Review result

The manuscript by Creon et al., entitled “Statistical crystallography reveals an allosteric network in SARS-CoV-2 Mpro”, presents a large-scale statistical analysis of 1,146 crystal structures of the SARS-CoV-2 main protease (Mpro). These structures are ligand-free and originate from fragment-based screening experiments, where such datasets are often not further analyzed once no bound fragment is identified. Here, the authors repurpose this otherwise underutilized structural information to identify correlated motions underlying allosteric regulation and enzymatic activation. The study examines structural correlations between the active site and Mpro residues and combines statistical crystallography with mutagenesis, crystallographic, and biophysical experiments to support the proposed allosteric network. The analysis is consistent with molecular dynamics (MD) simulations and shows similar covariance patterns around the native conformation, supporting the idea that the observed crystal structural variability reflects a subset of solution-phase conformational dynamics. Overall, this work will be of interest to crystallographers and structural biologists and is suitable for publication after addressing the comments below.

Comments

1. The paragraph beginning with ““These studies have identified four major sites as important for dimerization, shown in Fig. 1. First, a hydrogen bond between residues S1 and E166’, Finally, through specific point mutation or deletion of the C-terminal dimerization domain, monomeric variants of Mpro have been generated that show a collapsed oxyanion hole with a 310 helix spanning S139-N142.”

This provides a useful overview of the residues involved in Mpro dimerization. However, the text refers only generally to Fig. 1 without clearly indicating which specific panels of Fig. 1 correspond to each described interaction, making the description difficult to follow. This issue occurs not only here but also throughout the manuscript, where figures are frequently cited without specifying the relevant panels. Please indicate the appropriate figure panels when referring to specific structural features to improve clarity and readability.

2. Fig. 1A shows residues involved in Mpro dimerization. Besides the S139–N142 loop influenced by N214, are other dimerization-related residues also identified in the current covariance analysis?

3. The authors identify several residues proposed to exert allosteric effects on the active site, and these findings are reported to agree with MD simulations. However, the MD simulations were performed at temperatures higher than room temperature. Have these residues been compared with available room-temperature Mpro structures in the PDB? Do the correlated regions identified here agree with the conformational heterogeneity observed in those structures?

The manuscript states: “We then compared the results to a 100 μ s MD simulation of Mpro in solution conducted by D. E. Shaw Research.” Please provide a clear reference for this dataset (e.g., DOI or website link). In addition, reference 34 is missing a DOI.

4. As the analysis relies on a large dataset of 1,146 structures, it would be helpful to provide a summary of data quality, including resolution range, $\langle 1/\sigma(I) \rangle$, and CC1/2, either as tables or distribution plots. Data quality statistics should also be reported for the four datasets used in ensemble refinement.

5. The color schemes in Figs. 1A and C are difficult to distinguish, and the rotation between panels 1B and 1C is unclear. For Figs. 1D and 1E, the oxyanion loop (S139–C145) and N214 are not clearly shown together, making the interactions difficult to interpret. Finally, the figure legend should clearly explain all dash line colors or directly in figure; the magenta

color in Fig. 1G is not defined.

6. Fig. 2C shows a covariance pattern similar to that in Fig. 2A. Does this indicate that ligand-bound Mpro structures exhibit similar covariance to that observed in the ligand-free dataset of 1,146 structures? Have ligand-bound and ligand-free structures been directly compared to assess whether the covariance between N214 and the active site is maintained upon ligand binding? In particular, have the authors systematically examined whether the residues identified as allosterically coupled to the active site exhibit similar correlations in the full set of available ligand-bound Mpro structures?

7. In Fig. 5, the color scale used to represent map density does not appear to be consistent across panels (A, B and C are fuzzy). Please clarify whether identical contour levels and color mappings were used throughout.

8. The image quality in Fig. 6 is relatively low, making some structural details difficult to discern. Providing a higher-resolution version would improve clarity.

Version 1:

Reviewer comments:

Reviewer #1

(Remarks to the Author)

The authors have responded to the reviewer comments and revised the manuscript accordingly. Most concerns have been adequately addressed, and the revisions improve the clarity and presentation of the work.

Some points are only partially addressed, particularly the comparison between crystallographic covariance and molecular dynamics simulations and the potential influence of crystal packing. A brief clarification in the manuscript acknowledging that the comparison is largely qualitative, and that lattice contacts may contribute to the observed correlations, would further strengthen the interpretation.

Overall, the manuscript is scientifically sound and acceptable for publication after these minor clarifications.

Reviewer #2

(Remarks to the Author)

The authors addressed all the previous comments satisfactorily.

Reviewer #3

(Remarks to the Author)

The revised manuscript has significantly improved, and the authors have addressed my questions. I have no further comments.

made.

Dear Dr. Fontana,

Thank you for forwarding the reviews of our manuscript. Our team agrees that the reviewers made a number of important points. Addressing them has made the paper clearer and more rigorous. We'd like to thank the reviewers for providing constructive and actionable criticism, which is not always the case.

I attach our response to the reviews, where we address each of point in turn and highlight changes to the manuscript where appropriate. The original feedback from the reviewers is reproduced unabridged in **blue text**, with our responses interleaved; changes to the manuscript text are highlighted in **red**.

In addition, we have provided an updated version of the manuscript and SI in MS Word format with changes tracked. No major intellectual aspects of the paper have changed, but the language has gotten more precise and some new results are presented in the SI that the technical reader will find interesting.

We hope you find our response compelling and find our updated manuscript suitable for publication in Communications Biology. Please don't hesitate to contact me should the need arise.

Sincerely,

Thomas Lane

On behalf of the authors

Reviewer #1 (Remarks to the Author):

The manuscript “Statistical crystallography reveals an allosteric network in SARS-CoV-2 Mpro” by Creon et al. presents an exceptionally large crystallographic dataset and introduces an innovative framework for extracting covariance information across perturbed crystal structures. The study is technically impressive and conceptually novel. However, several interpretative and methodological points require clarification to ensure that the conclusions are strong and not overstated.

1. The manuscript introduces “statistical crystallography” as a general framework but does not clearly define its limits. The authors should explicitly state which types of motions are accessible (e.g., low-energy elastic deformations compatible with lattice integrity) and which are not, to avoid overgeneralization.

Good suggestion. We have added the following text in the discussion where we introduce the notion of statistical crystallography and discuss it precisely:

Along with our contribution, these studies demonstrate the crystal lattice is flexible enough to support continuous changes to the structure, and if subjected to enough random perturbations, can produce a conformational distribution. **Evidenced by these studies, this distribution can capture harmonic backbone displacements, side chain rotations, and near-native stable conformers compatible with the crystal lattice. Fluctuations that are either so large they disrupt the crystalline order or so small they cannot be seen at a given crystallographic resolution will be invisible.**

2. Although lattice effects are discussed, but more quantitative comparison between crystallographic covariance and MD simulations. Explicit metrics distinguishing shared versus crystal-specific covariance would reinforce the argument that the identified networks reflect intrinsic protein behavior.

When we were preparing our manuscript, we shared this desire to present a quantitative comparison between covariance matrices. Ideally, we could put a number on things: is our crystal covariance “close” to the MD covariance in some significant way? Or not? In short we agree totally with the desire of the reviewer for a quantitative comparison.

A moment’s reflection shows this isn’t a trivial question. How does one measure “close”ness? There are many metrics between matrices, each implying a different topology. Which to choose? And what value along that yardstick should we consider satisfactory? It’s clear that we need some theory, currently lacking and also beyond the scope of this paper, to definitively answer these questions.

While it may be challenging to define an absolute sense of closeness between two covariance analyses, a relative measure is much simpler – and we could compare, for instance, how close our crystal set is to an MD simulation vs. the distance between different simulations. We decided to pursue this pragmatic approach, which is reported in Supplemental Figure 5, reproduced below:

Supplemental Figure 5. Comparison to other MD simulations. Shown are covariance predictions for three distinct simulations of the wild-type M^{pro} dimer, (A) the 100 μs DESRES trajectory (6), presented in the main text (B) two 10 μs trajectories made available by RIKEN (7), and (C) 5 distinct 200 ns Gaussian accelerated simulations reported by Sztain et al. (GaMD) (8). All simulations were subsampled at 1 ns snapshot intervals. Panel (D) shows a quantitative heuristic of similarity between these covariance matrices and those from the crystallographic distributions (“DESY” corresponding to this work). Lower numbers and darker colors indicate more similar covariance matrices. Specifically, the pixel colors and numerical values report the Frobenius norm between the correlation-normalized covariance matrices. The comparison was only performed over a single monomer to facilitate comparison to the crystal structures, specifically residues 1-303 inclusive, as not all the Diamond crystals structures have the entire C-terminus modeled. The covariance models are generally in agreement, with the closest resemblance between the DESRES and GaMD simulations, followed by the crystallographic datasets, with the RIKEN simulation being the most distinct. Note that in addition to sampling, differences between the MD simulations may be due to the simulation configuration, with (A) the DESRES simulation based on the 6Y84 crystal structure, performed at 298 K with the DES-Amber force field, with His80 protonated at $N\delta$ and other histidines protonated at $N\epsilon$. In contrast, (B) the RIKEN simulation was started from the 6LU7 crystal structure, run at 310 K using Amber99sb-ildn, with His64 and His80 protonated at $N\delta$ and other histidines at $N\epsilon$ and (C) the GaMD simulations used the 6LU7 crystal structure, 310 K, Amber ff14SB force field, with His64, His80 and His164 protonated at $N\delta$ and all other histidines protonated at $N\epsilon$.

We note that while the different MD simulations are broadly in agreement, they do differ in meaningful ways. There are a number of choices that need to be made that can affect simulation results: starting model, force field, sampling technique, protonation states, *etc.* Therefore concrete points of comparison between simulation and experiments, such as ours, seem highly desirable to build consensus and credibility for both. We argue this in our Discussion, but wanted to re-iterate it here.

3. N214, Q256, and S284 are presented as equivalent covariance hotspots despite markedly different experimental outcomes. The authors should distinguish experimentally validated sites (e.g., N214) from putative or structurally tolerant sites and clarify that covariance alone does not necessarily imply functional sensitivity to alanine substitution.

This is a really excellent suggestion that helped clarify the narrative. In the “Mutational studies test predicted covariance hotspots” section, we modified the text to read:

Due to the strong correlation between N214/Q256/S284 and the active site, we predicted both effects would be observed. Therefore, we refer to these specific sites on the dimer interface, where structural change influences the dimer activity (and not just the concentration of dimers), as *putative covariance hotspots*.

We set out to test this hypothesis and chose to do so by making alanine substitutions at the N214, Q256, and S284 hotspot locations, then characterizing these mutants through a series of biophysical, biochemical and crystallographic experiments. *Because alanine substitution is specific perturbation that may or may not significantly disrupt the structure at the site of the mutation, if these alanine mutants disrupted the enzymatic function of M^{pro}, we elected to call them alanine-confirmed covariance hotspots. We note that if a specific alanine substitution does not perturb the enzyme structure significantly, we cannot definitively conclude the site is not a hotspot.*

And later in the text:

Based on our covariance model, we predicted our mutants would exhibit both a reduction in monomer/dimer affinity, and a reduction in activity of the residual dimers. Experiments on N214A confirmed this prediction; *this position is an alanine-confirmed allosteric hotspot.* Q256A shows a modest reduction in dimer and substrate binding affinities. Finally, mutating S284 to alanine failed to significantly perturb the protein structure, and the effect of perturbation at this position remains to be studied. *These sites remain putative covariance hotspots.*

4. In Supplemental Table 1, the S284A dataset shows an outer-shell $\langle I/\sigma(I) \rangle$ of 0.30 and CC1/2 of 0.179, values well below commonly accepted resolution cutoff thresholds. The manuscript does not justify inclusion of this shell. The authors should clarify the cutoff criteria used and indicate whether truncation at a more conservative threshold affects refinement statistics or conclusions regarding S284A.

Thank you for pointing that out. We have revised our refinement for the S284A mutant using a standardized cutoff of $CC_{1/2} > 0.3$. Table 1 in the supporting information has been revised accordingly.

5. The structural interpretation of Q256A remains ambiguous due to altered crystal packing. The authors should more clearly acknowledge that the observed rearrangements cannot be unambiguously attributed to intrinsic allosteric effects.

Absolutely true, and we do not wish to imply otherwise. To ensure this point is clear, we have amended the text as followed:

The site of mutation in our Q256A structure is directly involved in a new crystal contact, and there is no wild-type structure with the same contacts. Therefore, while the dimer interface is substantially perturbed in our Q256A structure, we cannot say if this is due to the novel crystal lattice, or an effect due to the mutation that would be retained for the protein in solution. Attempts to obtain crystals with alternative lattices have so far been unsuccessful. **The observed rearrangements cannot be unambiguously attributed to intrinsic allosteric effects. Our covariance analysis suggests that** Q256 sidechain plays a role in the structure of this important region of the protein that is disrupted upon mutation to alanine; at present, this remains conjecture.

6. The analysis is restricted to $C\alpha$ covariance, although key allosteric and catalytic interactions in Mpro are sidechain-mediated. The authors should explicitly justify this limitation and briefly discuss how sidechain contributions may be underrepresented.

Another good point. We added the following text to the Discussion section:

... our statistical work focused only on the covariance of $C\alpha$ atoms. Important sidechain rotations, sidechain-backbone coupling, and non-Gaussian displacements of the backbone remain to be analyzed (see Supplemental Figs. 2, 3, 15), **and for now preclude the discovery of allosteric interactions such as those mediated by R298 that involve sidechain motion without significant accompanying backbone displacement.**

We would also like to draw attention to the following paragraph in the Discussion, which focuses on this topic in the context of a specific important residue, R298:

Moreover, our present approach cannot comprehensively identify all residues known to form critical cross-dimer interactions. For example, R298 forms a salt bridge known to stabilize the Mpro dimer^{38,53} but does not appear prominently in our covariance analysis. A manual analysis of this residue shows that in our crystal's monoclinic lattice, the backbone of this residue is static, while the sidechain – which our analysis is not sensitive to – is only partially ordered (Supplemental Fig. 15). In other lattice packings, however, the sidechain becomes ordered, but the backbone varies from structure to structure. Therefore, to capture a more comprehensive set of covariance interactions, future efforts should study multiple crystal

packings and aim to develop algorithms that can efficiently capture side chain information.

7. Global Rwork/Rfree values do not guarantee local interpretability. Providing local map quality metrics (e.g., real-space correlation coefficients or representative density for key regions) would strengthen confidence that subtle structural differences are supported by experimental data.

Because our covariance analysis includes the entire structure of 1146 distinct crystallographic datasets, we felt a statistical response (vs. cherry picking density examples) was warranted. Therefore, we added a new Supplemental Figure, numbered 1 in the revised SI and reproduced here for completeness:

Supplemental Figure 1. Overview of the data quality of the 1146 structure dataset. (A) Real-space correlation between the $2mF_o - DF_c$ map and the model (F_c) for the set of structures presented as a function of residue (all atoms in a residue averaged). The mean value is shown in blue, while the extrema across all structures is shown in grey. **(B)** Histograms showing the distribution of dataset resolution limits, determined as where $CC_{1/2}$ falls below 0.5, and **(C)** refinement R-factors.

8. The N214A mutant provides strong validation, but the mechanistic link between N214 disruption, N-terminal disorder, oxanion hole destabilization, and reduced catalysis could be articulated more explicitly.

Thank you for bringing this to our attention. To clarify, we have revised the paragraph on the N214A mutant, reproduced below:

Protomer B of our structure shows this disruption most prominently. Specifically, lacking the stabilizing interactions provided by N214, the N-terminus shifts, breaking hydrogen bonds with E166' that hold the N-terminus in place (Fig. 5C). This disrupts a series of hydrogen

bonds between the oxyanion loop and N-terminus that normally stabilize the catalytic conformation (Fig. 5A and C). The loss of the carboxamide sidechain at N214 precludes a hydrogen bond with G2 in the N-terminus of the same protomer. This results in a flip of the amino acids F3, S139' and F140', the disruption of additional hydrogen bonds between S1 and the backbone of P140' and S1 and E166', the formation of new hydrogen bonds between S1 and F3 and G2 and S139'. The loss of the hydrogen bond network leads to shifts in the atomic positions of C143, G144, and C145 (SI Fig. 13A). The mean position of the oxyanion hole on the dimeric protomer shifts, even though the hydrogen bond between S144' and L141' persists (Supplemental Fig. 13). Additionally, a hydrogen bond between Q299 and S139' (Fig. 5A(ii)) is exchanged for one between S139' and N-terminal residue G2, resulting in a disordered C-terminus (residues 301-306). Without these stabilizing interactions that bridge the two protomers, the oxyanion loop collapses. In contrast to the monomeric structure M_{pro}1-199 construct (PDB: 7UJ9), in our N214A structure the loop formed by residues S139 to L141 does not form a 310 helix. Instead, we detected a loss in observable density and shift in mean position of residues L141 to G143, potentially reflecting an intermediate state between the monomer construct and wild type. (Fig. 5A(i) and Supplemental Fig. 13). Dimerization and catalytic turnover are compromised as a result, confirming N214 is an alanine-confirmed hotspot and plays a pivotal role in the allosteric network that connects dimerization to function in M^{pro}.

Reviewer #2 (Remarks to the Author):

The manuscript by Creon et al. builds on the authors' previous work where they performed crystallographic ligand screening, producing, among many ligand-bound SARS-CoV-2 MPro structures, a very large set of apo-structures. This set has now been utilized to down-select 1146 structures for structural analysis using the atomic displacement covariance as a measurable change in the enzyme conformations. This method, introduced as statistical crystallography, identified three covariance hotspots in the MPro sequence – residues that may be allosterically linked to the active site and influence the enzyme activity. Three point mutations have been introduced and the activity, dimerization and structures of N214A, Q256A and S284A mutants have been studied. Interestingly, only N214A mutation demonstrated a significant effect on the enzyme dimerization (Kdimer), its activity and structure. This is an interesting result, but the presentation, interpretation and conclusions need to be improved and checked for factual accuracy.

The major issue is with the subsection on Variant N214A. The leading paragraph states that N214 side chain bridges the dimer interface by interacting with the N-terminus of the dimeric protomer. This statement is confusing because N214 H bonds with the main chain of G2 of the same protomer. Then, how does it bridge the dimer interface? Figure 5 does not explain this at all.

Thank you for pointing this out. We optimized our text and Figure 5 (reproduced below) accordingly to make it more visible how that network interacts. Based on reviewer #1, point 8, we have already amended the text and clearly emphasized that it is only the hydrogen bond network that creates the bridge between the two protomers and not the side chain of N214 itself. Figure 5C and the figure captions have been optimized accordingly and the hydrogen bonds have been added.

Figure 5. Structures of the N214A, Q256A, and S284A mutants. Mutant structures are shown in blue (chain A) and orange (chain B) with a wild-type structure (light grey) as a reference. Two wild-type structures were employed in order to match the space group and lattice parameters between the reference and mutant structures as closely as possible, specifically PDB: 7BB2 for the N214A (PDB: 9GI6) and Q256A (PDB: 9GHN) mutants (P2₁,2₁,2₁) and PDB: 7AR6 for the S284A (PDB: 9GHO) mutant (C2). All densities shown are 2mF_o-DF_c maps at 1 RMSD. (A) Notable changes in the N214A mutant crystal structure (PDB: 9GI6). (i) Removal of the N214 sidechain disrupts a hydrogen bonding network (which is shown in panel C), displacing F140' and S139' which form part of the oxyanion loop in the active site of the dimeric protomer. Shown in cyan is the structure of bound inhibitor (GC376, PDB: 7D1M) also shown in Fig. 1 as a reference. (ii) This displacement further disrupts a hydrogen bonding interaction between S139' and Q299 (grey). Q299 is located in an α -helix close to the C-terminus, which is displaced from the dimer interface. (B) Notable changes in the Q256A mutant structure (PDB: 9GHN). Upon mutation, two new crystal contacts form. (i) The first contact occurs due to the flip of a β -hairpin (D153-C156), which inserts the phenol sidechain of Y154 into a pocket formed by N51*-N53* of an adjacent dimer, an interaction stabilized by hydrogen bonds between these residues (asterisks denote distinct symmetry-related units in the crystal). (ii) In the second contact, the α -helix comprising amino acids L227** to Y237** from a second neighboring dimer fill a hydrophobic pocket generated by the removal of the Q256 sidechain. (iii) This results in a new crystal lattice, distinct from (iv) the wild type. Moreover, (v) the interaction between S1 and E166' is impacted by a displacement of the α -helix containing N214. This movement is accompanied by a small shift in the position of S1, increasing its distance from E166' and weakening their interaction. (vi) Consequently, the C-terminus is displaced. (C-E) Details of the variant structures at the site of mutation, with the mutated residues highlighted in purple. (C) **Changes in the hydrogen bond network in the N214A mutant.** The loss of the carboxamide sidechain at N214 precludes a stabilizing interaction with G2 in the N-terminus of the same protomer. This results in the flip of F3, F140', and S139', the disruption of hydrogen bonds between S1 and the backbone of P140' and S1 and E166', the formation of new hydrogen bonds between S1 and F3 and G2 and S139'. The loss of the hydrogen bond network leads to shifts in the atomic positions of C143, G144, and C145. The oxyanion hole on the dimeric protomer shifts, even though the hydrogen bond between S144 and L141 still exists (Supplemental Fig. 13). (D) At the site of mutation, swapping Q256 to alanine results in the displacement of the C-terminus. (PDB: 9GHN, blue / PDB: 7BB2, grey). (E) The hydrophobic zipper (magenta) at the dimer interface that contains S284 is nearly unchanged upon mutation of this residue

to alanine. Only an elongation of the inter-protomer distance from 4.8 to 5.5 Å of the (A285-A285' C_α, green) was observed (PDB: 9GHO, blue / PDB: 7AR6, grey).

The text then goes on to talk about how N214A mutation disrupts the oxyanion hole, which made of residues 143, 144 and 145, not 139 through 145 as the authors claim, because only 3 amides can create an oxyanion hole. Here, again, if one double checks the structure, it becomes clear that N214 is 14A away from G143, so that it is hard to visualize how such a distant residue can disrupt the oxyanion hole. Thus, the H bonding network needs to be depicted for both the wt and N214A. Has the oxyanion hole actually unwound like in the monomeric structures (e.g. MPro1-199)?

Good point that we could be more precise. We have changed the relevant wording to make it clear whether we are referring to the oxyanion hole or loop that contains the precise oxyanion hole residues:

Our structure of N214A provides insight into why the loss of the N214 sidechain disrupts dimerization and catalysis so dramatically. M^{pro} employs an oxyanion hole (G143 to C145), embedded in a loop (S139 to C145), which we refer to as the oxyanion loop, to stabilize the reaction transition state highlighted in structures of transition state analogs (Fig. 1D). In the wild-type system, the sidechain of N214 participates in a hydrogen-bonding network mediated by the N-terminus that stabilizes this oxyanion loop (Fig. 1E). Recent MD simulations pinpointed the N-terminus as a key mediator of the cooperativity between the two dimeric active sites⁴². In our N214A structure, in which the protein forms a native dimer, this network is disrupted and the oxyanion loop is only partially ordered (Fig. 5A(i) and C, Supplemental Fig. 13). Since ITC and nDSF measurements show N214A is a competent substrate binder, we propose that removal of the N214 sidechain reduces the population of conformations capable of stabilizing the reaction transition state, in turn compromising enzyme activity.

We added the hydrogen bond network in both Figure 1 and Figure 5 and adjusted the corresponding figure captions.

Figure 1. Structure of the M^{pro} homodimer. Dimerization is obligate for M^{pro} enzyme turnover. **(A)** Through mutagenesis studies, several residues on the dimer interface have been identified as essential for the formation or function of the dimer assembly. These sites on the dimer interface are generally distal from the active site (teal). As part of this work, we identified three residues (purple) that are predicted to have strong structural correlation with the active site. **(B)** The dimer assembly. **(C)** Highlighted are the three residues we mutated in this work (purple). N214 and S284 are located at the dimer interface, while the final residue, Q256, is solvent exposed. **(D)** The active site oxyanion hole (G143, S144 and C145) is formed by backbone amide groups along the oxyanion loop (S139-C145). The ability of this structure to stabilize a tetrahedral transition state is highlighted in structures of transition state analogs, such as the covalent inhibitor GC376, shown here forming a hemithioacetal linkage (PDB 7D1M). The hydroxyl portion of the hemithioacetal is disordered, and sits in two positions that approximate the tetrahedral transition state geometry. **(E)** N214 (purple) forms a hydrogen bonding network with the N-terminus, which in turn interacts strongly with a loop (S139 to C145) that contains the catalytically essential oxyanion hole in the active site and harbors the central catalytic thiol. **(F)** Q256 (purple) is largely solvent exposed, but is adjacent to the C-terminus, which is stabilized by hydrogen bonds between the Q256 sidechain and S301 backbone. Finally, **(G)** S284 (purple) forms part of a hydrophobic zipper (included interactions are shown in magenta) that bridges the dimerization domains of both dimer-forming protomers. The distance of 4.8 Å between the C_{α} atoms of A285 and A285' is highlighted in green. Hydrogen bonds are shown in yellow (2.2 to 3.5 Å). All panels show PDB 7BB26868, except (D), which shows 7D1M8686.

While it repeats material found in our paper, we wanted to address the reviewer's specific structural questions here. The loop containing the oxyanion hole does not unwind as it does in the monomeric structure – we do not see the helix formation (see SI Fig 13). Instead, there is an apparent destabilization due to loss of favorable interactions. This is reflected in a decrease in clear $2mF_o - DF_c$ density that indicates increased flexibility. We tried to fit in the monomeric structure early in the beginning of the refinement of the N214A mutant, but this conformation is not consistent with the

crystallographic data. While speculative, we think that our N214A structure is in an intermediate conformation in-between the monomer and the dimer conformations already found in the PDB.

It is claimed that the oxyanion hole collapses – what does that mean?

Thanks for pointing out that this language is not precise. We have elaborated, writing:

Without these stabilizing interactions that bridge the two protomers, the oxyanion loop collapses, **adopting a partially disordered conformation in which the C143, G144 and C145 amides are not oriented to support catalytic activity.**

Further, SI Fig. 13 shows the conformations visually and explicitly.

The quality of the depicted electron density in Figure 5A and 5C is poor. Make the density more visible. In Figure 5C, does the N-terminus even have interpretable density?

Thank you for this feedback. We improved Figure 5: the density is now clearer in panel 5A, C and also in B(v). We can see density for the N-terminus after amino acid F3. We see clear density for the aromatic ring, which undergoes a flip into solvent as a result of the mutation of N214 into an alanine.

Density for S1 and G2 is not visible and hard to interpret. Nevertheless, due to the density in the residue and the backbone for F3 we can conclude that the backbone molecules of the two first amino acids have to shift closer to the N214A position. The loss of clear density indicates a higher flexibility of the N-terminus in the N214A variance, which makes sense as G2 is stabilized by N214 in the wild-type M^{Pro}.

The first paragraph of subsection Variant Q256A states that the C-terminus plays a significant role in forming the dimer complex. However, in many MPro structures the C-terminal residues beyond 302 are disordered, with sometime no detectable electron density. That means the C-terminus is disordered, supported by numerous MD simulations. How can it play a significant role in dimer formation?

We want to thank the reviewer – this statement was not precisely formulated and indeed is simply not correct as stated. We simply meant to state that the C-terminus sits between the two protomers. Upon reflection, we simply deleted this statement, as any relationship between the C-terminus and Q256 is speculative and not immediately relevant.

Thanks to the reviewer for this distinct improvement to the text.

The subsection “Three variants yield a single allosteric network” is rather dubious. How was the prediction made that mutants would exhibit reduction in activity – just covariance plots do not predict that. What is a single allosteric network? Either, rewrite this section to explain what this is about, or completely remove it.

Thanks for this comment, as indeed after the reviewer pointed it out, we agree that the title of this

section was simply bad. We've replaced it with "Of three alanine mutants, one is a confirmed covariance hotspot", which we hope the reviewer will agree is an improvement. The "confirmed" terminology follows our response to the suggestion of Reviewer 1 (see above).

The last paragraph before Discussion describes a very confusing, strange and poorly articulated exercise, including Figures 6 and S14. How were the crystals exposed to air – sitting in their drops for a while, right after mounting on a loop, after cryoprotecting? What were the crystal drop volumes for these experiments? Methods section doesn't appear to have any details. It is also hard to believe that drops of several microL in volume would lose significant amounts of water in just seconds to be observed in the unit cell contraction.

Thanks for pointing out that this was confusing and required additional explanation, an oversight on our part. We will reply to the reviewer's questions by highlighting specific additions to the manuscript that clarify the experimental procedure. In the last paragraph before the Discussion, we added:

To test this hypothesis, we opened a single vapor diffusion well to the dry lab atmosphere and fished multiple crystals as a function of time (Supplemental Fig. 14). Crystals were therefore able to dehydrate in situ in a 0.5 μ L (set volume) sitting drop and fished at intervals spanning 5 to 205 seconds. Crystals fished at longer delays failed to diffract to high resolution.

and to the Methods, we added a new paragraph:

For crystals fished as a function of time, we reproduced the crystallization conditions used in our ligand screen³¹ but omitted any ligands. Specifically, crystallization was achieved by mixing 0.23 μ L of protein solution (6.25 mg/mL) in 20 mM HEPES buffer (pH 7.8) containing 1 mM DTT/TCEP (respectively), 1 mM EDTA, and 150 mM NaCl with 0.22 μ L of reservoir solution consisting of 100 mM MIB, pH 7.5, containing 25% (w/w) PEG 1500 and 5% (v/v) DMSO, and 0.05 μ L of a micro-seed crystal suspension using an Oryx4 pipetting robot (Douglas Instruments). This growth solution was equilibrated by sitting drop vapor diffusion against 40 μ L reservoir solution. A timer was set and crystals were manually harvested and cryo-cooled in liquid nitrogen at specified intervals for subsequent X-ray diffraction collection.

The intent was to replicate the conditions used in the ligand screen as closely as possible. Note the small drop volume (500 nL) employed.

Reviewer #3 (Remarks to the Author):

Review result

The manuscript by Creon et al., entitled “Statistical crystallography reveals an allosteric network in SARS-CoV-2 Mpro”, presents a large-scale statistical analysis of 1,146 crystal structures of the SARS-CoV-2 main protease (Mpro). These structures are ligand-free and originate from fragment-based screening experiments, where such datasets are often not further analyzed once no bound fragment is identified. Here, the authors repurpose this otherwise underutilized structural information to identify correlated motions underlying allosteric regulation and enzymatic activation. The study examines structural correlations between the active site and Mpro residues and combines statistical crystallography with mutagenesis, crystallographic, and biophysical experiments to support the proposed allosteric network. The analysis is consistent with molecular dynamics (MD) simulations and shows similar covariance patterns around the native conformation, supporting the idea that the observed crystal structural variability reflects a subset of solution-phase conformational dynamics. Overall, this work will be of interest to crystallographers and structural biologists and is suitable for publication after addressing the comments below.

Comments

1. The paragraph beginning with ““These studies have identified four major sites as important for dimerization, shown in Fig. 1. First, a hydrogen bond between residues S1 and E166’, Finally, through specific point mutation or deletion of the C-terminal dimerization domain, monomeric variants of Mpro have been generated that show a collapsed oxyanion hole with a 310 helix spanning S139-N142.” This provides a useful overview of the residues involved in Mpro dimerization. However, the text refers only generally to Fig. 1 without clearly indicating which specific panels of Fig. 1 correspond to each described interaction, making the description difficult to follow. This issue occurs not only here but also throughout the manuscript, where figures are frequently cited without specifying the relevant panels. Please indicate the appropriate figure panels when referring to specific structural features to improve clarity and readability.

Thank you for pointing this out. We revised our manuscript and added the specific panel numbers to all cited figures.

2. Fig. 1A shows residues involved in Mpro dimerization. Besides the S139–N142 loop influenced by N214, are other dimerization-related residues also identified in the current covariance analysis?

We focused on the three most surprising correlations we discovered, but indeed many residues that are more trivially adjacent to the active site but that bridge the dimer interface are involved. We’ve elaborated on this in the Results, writing:

Proceeding with the backbone analysis, three motifs of highly covarying substructures emerged. The first were mobile loops and surface-exposed amino acids directly proximal or contiguous with the active site, **many of which bridge the dimer interface, most notably the N-terminus**. More surprisingly, the second group contained a set of residues that form crystal contacts. The final category represented three regions of the dimerization domain, with high covariance peaking at N214, Q256, and S284 (Fig. 3), highlighting that these locations are

structurally coupled to the active site.

But we do not wish to claim that our method, especially the current C_{α} backbone analysis, can capture all the important interactions and networks. In the Discussion, we write:

Moreover, our present approach cannot comprehensively identify all residues known to form critical cross-dimer interactions. For example, R298 forms a salt bridge known to stabilize the Mpro dimer^{38,53} but does not appear prominently in our covariance analysis.

3. The authors identify several residues proposed to exert allosteric effects on the active site, and these findings are reported to agree with MD simulations. However, the MD simulations were performed at temperatures higher than room temperature. Have these residues been compared with available room-temperature Mpro structures in the PDB? Do the correlated regions identified here agree with the conformational heterogeneity observed in those structures?

We have not conducted an extensive analysis of room temperature M^{pro} structures, nor structures under a number of other possible conditions: different crystal systems (pH, spacegroups/packing, salts, etc) that can affect structure. Each is interesting, but to estimate a covariance matrix robustly requires 100 or more structures. Therefore, at present we have no strong basis on which to perform a room temperature comparison.

We do believe such a comparison would be interesting, but it is not trivial. As pointed out in the Discussion, cryo-freezing is likely a large perturbation that produces the distribution we see. Room temperature structures are likely to be significantly more homogenous to start – a different perturbation is likely to be helpful in understanding how the room temperature structure covaries. In the Discussion, we write:

... serial crystallography at room temperature provides an established and facile route to the study of up to millions of crystals in a single experiment⁵⁶. In the presence of perturbations due to external fields⁵, humidity⁵⁷, temperature⁵⁴, or acoustic waves⁵⁸, room-temperature serial experiments could provide a second route to routine statistical crystallography.

Finally, we point out that any crystal system, at any temperature, is obviously different from the protein in solution. This is tackled extensively in the Discussion, the most important part being where we write:

For instance, we highlighted similarities between our crystallographic data and molecular dynamics simulation as a point of strength. MD simulations can model proteins in solution, without the potential caveat of restrictions due to lattice confinement. On the other hand, MD relies on classical force fields that may introduce error and may be limited by sampling in the cases of large systems or many variants of interest (mutants, ligands, etc.). We analyzed three distinct MD simulations of Mpro that had been previously published, and found that while broadly similar, the predicted atomic covariances differed between them in meaningful ways (Supplementary Figs. 5 and 7). In light of these important differences, experimental techniques such as statistical crystallography – as well as established NMR and emerging

cryoEM methods to access distributions of structure – are a necessary complement to simulations.

The manuscript states: “We then compared the results to a 100 μ s MD simulation of Mpro in solution conducted by D. E. Shaw Research.” Please provide a clear reference for this dataset (e.g., DOI or website link). In addition, reference 34 is missing a DOI.

Thanks for pointing this out, as it’s critical that others be able to link to these simulations and access them as well. If our paper is accepted, we will be sure to work with the editorial staff to correctly reference and cross-link these datasets.

Unfortunately, to the best of our knowledge, there is no DOI or clear reference for the DESRES dataset. It was available for download via the DESRES website during the pandemic, with instructions to cite the work exactly as we have. That original page has ceased to exist; now, there is a permanent copy available via the COVID MOLSSI simulation archive:

<https://covid.molssi.org/simulations/#desres-100-%c2%b5s-md-of-3clpro-no-water-or-ions>

But no associated DOI to the best of our knowledge. For Ref 34, added the appropriate DOI (10.17632/vpps4vhryg.2/).

Again, if accepted, we will be sure to work with the editorial staff to work out what the best course of action is here, as we need some advice.

4. As the analysis relies on a large dataset of 1,146 structures, it would be helpful to provide a summary of data quality, including resolution range, $\langle I/\sigma(I) \rangle$, and CC1/2, either as tables or distribution plots. Data quality statistics should also be reported for the four datasets used in ensemble refinement.

We have included a new SI Figure 1 that provides a summary of the dataset quality, including real space CC values, resolution, and R-factors. See our reply to reviewer #1, point 7. Further, we expanded Table S2 to include resolution information for the ensemble refinements, which includes model statistics for these refinements. Note that we plan to make the ensemble refinements – as well as the refinements for all 1146 single-structure models in our crystal set – available via Zenodo for download, inspection, and any further analysis should our paper be accepted.

5. The color schemes in Figs. 1A and C are difficult to distinguish, and the rotation between panels 1B and 1C is unclear. For Figs. 1D and 1E, the oxyanion loop (S139–C145) and N214 are not clearly shown together, making the interactions difficult to interpret. Finally, the figure legend should clearly explain all dash line colors or directly in figure; the magenta color in Fig. 1G is not defined.

Thank you for these findings. We changed the color scheme in panel A and increased the stick radius of the N, Q and S residues in panel B and C. We hope the rotation with the optimized arrow helps to understand the rotation more easily. We optimized panel E by adding the hydrogen bond network to the structure and show the N214 position in combination with the oxyanion loop and the

N-terminus. We have chosen not to add N214 in panel D, as we show the hydrogen bond relationship in panel E. Panel D serves as an illustration of the oxyanion hole and its ability to stabilize a tetrahedral transition state. The magenta color is now defined in the figure caption for panel G.

6. Fig. 2C shows a covariance pattern similar to that in Fig. 2A. Does this indicate that ligand-bound Mpro structures exhibit similar covariance to that observed in the ligand-free dataset of 1,146 structures?

Yes, precisely. We state explicitly in the paper:

To evaluate how robust this finding was, we compared our set of crystals to a publicly released set of 95 M^{pro} structures from Diamond Light Source³⁵. These structures were refined by hand and bound to fragments, in contrast to our ligand-free automatically refined models. The Diamond data, containing an order of magnitude fewer structures, has a considerably weaker signal-to-noise ratio. Nonetheless, at the level of C_α covariance the two sets of crystals show qualitative agreement (Fig. 2).

Have ligand-bound and ligand-free structures been directly compared to assess whether the covariance between N214 and the active site is maintained upon ligand binding? In particular, have the authors systematically examined whether the residues identified as allosterically coupled to the active site exhibit similar correlations in the full set of available ligand-bound Mpro structures?

It's a good but broad question. Our ligand screen (Günther *et al*, Science 2021) only uncovered 37 unambiguously ligand-bound structures, and is therefore statistically underpowered. Therefore we elected to compare directly to the Diamond screen which contained 95 structures, which in addition to being larger had the added benefit of being collected and analyzed by a completely independent group.

As discussed, while the statistical power of this dataset is still low, the same overall covariance pattern is observed. We speculate that ligand binding in general will act as a further perturbation, on top of any heterogeneity induced by sample preparation, but we feel we cannot make a definitive claim about that with these data at this time.

Further, we felt the fact that covariance information could be obtained *without* any known ligands was an important intellectual contribution, and so centered our analysis around the ligand-free data. A full investigation of the effects of ligand binding on the covariance structure is a large topic which we hope to attack in future work.

7. In Fig. 5, the color scale used to represent map density does not appear to be consistent across panels (A, B and C are fuzzy). Please clarify whether identical contour levels and color mappings were used throughout.

That is true, thank you for this feedback. Reviewer 2 made the same point – see above. We have now new Figure 5 panels with updated map density.

8. The image quality in Fig. 6 is relatively low, making some structural details difficult to discern. Providing a higher-resolution version would improve clarity.

Thanks for pointing this out. We have higher resolution source materials for each panel and have updated the figure with a higher-resolution version, which we hope makes it through to the reviewer. Should our paper be accepted we will be sure to work with the editorial staff to ensure that all figures are drawn to a high quality in the published issue.